# CoSteer: Collaborative Decoding-Time Personalization via Local Delta Steering

## Abstract

Personalized text generation has become crucial for adapting language models to diverse and evolving users' personal context across cultural, temporal, and contextual dimensions. While existing methods often rely on centralized fine-tuning or static preference alignment, they struggle to achieve real-time adaptation under resource constraints inherent to personal devices. This limitation creates a dilemma: large cloud-based models lack access to localized user-specific information, while small on-device models cannot match the generation quality of their cloud counterparts. To address this dichotomy, we present **CoSteer**, a novel collaborative framework that enables decoding-time personalization through localized delta steering. Our key insight lies in leveraging the logits difference between personal context-aware and -agnostic outputs from local small models as steering signals for cloud-based LLMs. Specifically, we formulate token-level optimization as an online learning problem, where local delta vectors dynamically adjust the remote LLM's logits within the on-device environment. This approach preserves privacy by transmitting only the final steered tokens rather than raw data or intermediate vectors, while maintaining cloud-based LLMs' general capabilities without fine-tuning. Through comprehensive experiments on various personalized generation tasks, we demonstrate that CoSteer effectively assists LLMs in generating personalized content by leveraging locally stored user profiles and histories, ensuring privacy preservation through on-device data processing while maintaining acceptable computational overhead. Our anonymized code and data are available at `https://anonymous.4open.science/r/Costeer-4977`

## 1 Introduction

The rapid development of large language models (LLMs) has significantly enhanced natural language processing, enabling these models to understand context and generate coherent text effectively (Zhao et al., 2023). This progress has sparked interest in personalization, where AI systems move beyond generic content to tailor interactions based on individual user profiles. Personalized generation refers to creating content customized through analysis of user-specific attributes like linguistic patterns, interaction histories, and contextual preferences (Xu et al., 2025). This approach enables user-specific outputs that maintain contextual relevance, with applications in personalized recommender systems (Lyu et al., 2023; Zhang et al., 2024a), adaptive dialogue agents (Wu et al., 2025), and customized content creation platforms (Mysore et al., 2024).

Existing personalized generation approaches primarily fall into two paradigms. The first involves training-based methods, which leverage user data to tailor models using two main strategies. Parameter-efficient architectures, such as PLoRA (Zhang et al., 2025a), employ a shared LoRA module across users to strike a balance between personalization and resource efficiency. In contrast, individualized adapters, as seen in UserAdapter (Zhong et al., 2021), optimize for per-user customization, albeit at a higher computational cost. Additionally, multi-objective reinforcement learning frameworks (Zhou et al., 2024; Wu et al., 2023) redefine personalization as the alignment of generic model capabilities with user-specific patterns, achieved through reward shaping. The second paradigm comprises tuning-free methods, which avoid model updates by adapting to user context during inference. This approach is further divided into two techniques. Prompt engineering implicitly encodes user profiles through curated prompts. For example, Cue-CoT (Wang et al., 2023a) integrates user-specific reasoning into system prompts, while PAG (Richardson et al., 2023) retrieves historical

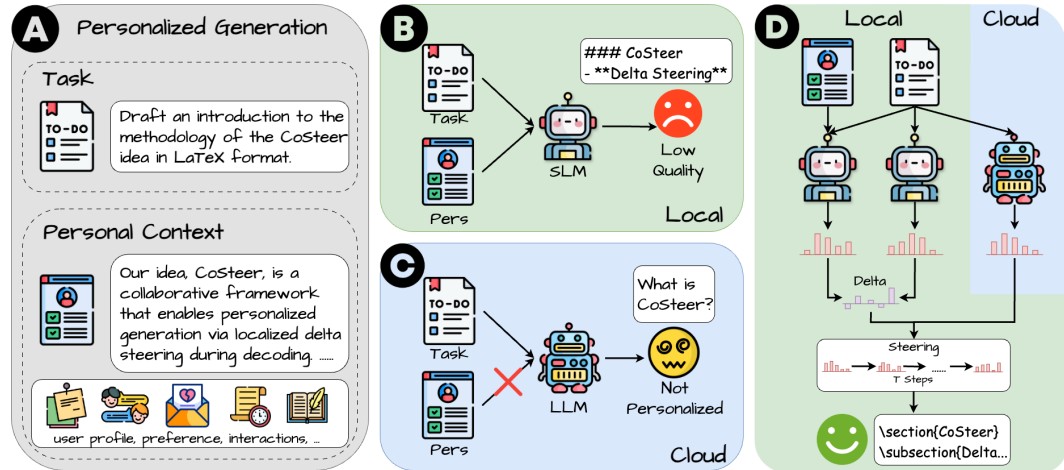

Figure 1: Schematic illustration of CoSteer framework. (a) Task scenario: A user poses a question potentially requiring access to local personal context (e.g., user profile, interaction history). (b) Limitations of small locally-deployed language models: Direct inference with constrained model capability leads to suboptimal generation quality. (c) Challenges of cloud-based LLMs: Despite strong generalization, once LLMs are constrained from accessing local personal context, they result in misaligned or contextually disconnected outputs. (d) CoSteer: Optimizes LLM predictions through local delta steering, effectively balancing the LLM's broad knowledge with user-specific information.

interactions to construct context-aware prompts. On the other hand, inference-time optimization techniques, exemplified by PAD (Chen et al., 2025) and Drift (Kim et al., 2025), actively adjust token generation probabilities by manipulating logit differences. Unlike prompt-based methods that inject semantic context, these techniques constrain output distributions to better align with user preferences.

However, both training-based and tuning-free methods face critical dilemmas in balancing privacy and quality when deployed on resource-constrained personal devices. Training-based methods require computational resources that may not be available locally, and cloud-based training risks exposing user data. Meanwhile, tuning free methods that transmit raw personal context to cloud-based LLMs for context-aware adaptation also risk exposing sensitive information, whereas relying solely on local SLMs leads to degraded output quality. Therefore, there is a pressing need to design a new framework that balances quality and privacy by combining the general capabilities of remote LLMs with the benefits of local information. Achieving this balance on personal devices faces two major challenges: 1) User information changes in real-time, necessitating methods for real-time adjustments to maintain personalization effectively. 2) To avoid the risk of user information leakage, any privacy-sensitive operations must be conducted locally.

To address these fundamental challenges, we propose **CoSteer**, a collaborative framework enabling real-time personalized generation via **localized delta steering** during decoding. Our core innovation resides in utilizing the logits difference, i.e., delta, between personal context-aware and -agnostic outputs from on-device small language models (SLMs) as dynamic steering signals to guide cloud-based LLM distributions. Specifically, the SLM generates two contrasting predictions: 1) personalized outputs incorporating privacy-sensitive personal context, and 2) generic outputs using users' queries only. The resultant logit differentials serve as indicators of personalization directions.

By formulating token-level adaptation as an online learning process, CoSteer establishes an iterative refinement mechanism that enables edge devices to locally optimize cloud-based LLM predictions, eliminating the need to transmit raw personal context or intermediate representations. Experimental evaluations demonstrate that CoSteer achieves superior performance compared to two conventional baselines: 1) cloud-based LLMs processing queries without personalized context, and 2) local SLMs conducting personalization context-aware inference independently. Remarkably, the framework attains comparable performance to the near-upper-bound baseline of cloud-based LLMs operating with full personal context access, while maintaining strict data isolation.

To sum up, our contributions are threefold:

- **Pioneering Focus on Personalized Generation in Resource-Constrained Environments**: We are among the first to address the issue of personalized generation in edge environments with limited resources, a problem of urgent research significance in today's context.

- **Introduction of the CoSteer Framework**: We propose the CoSteer framework, which optimizes LLM's predictions using local delta signals. This approach effectively combines the general capabilities of LLMs with the privacy benefits of on-device SLMs.

- **Comprehensive and Realistic Validation**: We provide substantial experimental evidence demonstrating the effectiveness of our proposed method. Our evaluation further extends to address critical real-world challenges, confirming the framework's practical value.

## 2 RELATED WORK

**Training-time personalization**    Current mainstream approaches to personalization predominantly focus on training-time adaptation, leveraging user-specific data to tailor models through parameter-efficient fine-tuning (PEFT) or reinforcement learning-based alignment. A subset of work explores one-PEFT-for-all-users strategies (Zhang et al., 2025a; Woźniak et al., 2024; Zhu et al., 2024; Kong et al., 2025), where a shared set of lightweight parameters (e.g., adapters or conditional batch normalization modules) is optimized to generalize across diverse user preferences. Conversely, the one-PEFT-per-user paradigm prioritizes individualized tuning through isolated parameter sets, enhancing personalization while preserving privacy by avoiding cross-user data leakage (Zhong et al., 2021; Peng et al., 2024; Tan et al., 2025; 2024). Beyond PEFT, RL-based approaches have gained traction for aligning models with user preferences through reward-driven optimization. Recent work has further formalized personalization as a multi-objective reinforcement learning (MORL) problem: methods like MORLHF (Wu et al., 2023) and MODPO (Zhou et al., 2024) train distinct reward models for different objectives and merge them during optimization, while alternative frameworks such as Personalized Soups (Jang et al., 2023), Reward Soups (Ramé et al., 2023), MOD (Shi et al., 2024), and PAD (Chen et al., 2025) dynamically combine policies from multiple trained models during the decoding phase. This series of work requires significant computational resources and relies on static datasets, which cannot meet the real-time personalization needs of users.

**Inference-time adaptation**    Methods like Linear Alignment (Gao et al., 2024) and Contrastive Decoding (Li et al., 2023) pioneer logit steering mechanisms that dynamically adjust token distributions during generation. By computing differential signals between personalized and generic outputs on edge devices, these approaches eliminate dependency on retraining while preserving privacy through localized computation. CoS (He et al., 2025) follows this approach by using an adaptable parameter $\lambda$ to scale the impact of personal information, thereby achieving controllable personalization. Amulet (Zhang et al., 2025b) uses online learning algorithms to iteratively optimize the logits distribution, precisely aligning with the user's preferences. However, this series of methods requires explicitly transmitting user information to the LLM, which poses a risk of privacy leakage (Yan et al., 2025) and hinders the practical deployment of these methods.

**Collaborative generation**    The computational constraints of edge devices have driven innovative paradigms in collaborative generation between cloud-based LLMs and local small language models. Existing collaborative generation approaches predominantly focus on two technical routes: (1) assistive alignment, where local SLMs are trained to augment the LLM's user preference alignment during inference, and (2) reward-guided decoding, where lightweight reward models provide preference signals through reward-guided decoding. For instance, Aligner (Ji et al., 2024) leverages natural language feedback generated by the SLM to directly inject user-specific linguistic patterns into the LLM's decoding process, while Expo (Zheng et al., 2025) achieves preference alignment through linear interpolation of layer-wise weights between LLMs and SLMs. Recent advancements further demonstrate that training specialized small reward models can effectively steer LLM outputs toward personalized objectives (Snell et al., 2024; Liu et al., 2025). Among these, Proxy-tuning (Liu et al., 2024) and Cogensis (Zhang et al., 2024b) are closest to our design. Proxy-tuning modifies the LLM's logits distribution by contrasting pre-trained and fine-tuned SLM outputs, whereas Cogensis employs a learned fusion network to combine logits from both models. In contrast to these existing methods, our proposed CoSteer framework establishes a completely tuning-free collaboration mechanism, eliminating the need for training.

To clearly illustrate the technical positioning of our work, we have created Table 3 to highlight the key differences between our approach and related studies.

# 3 METHODOLOGY

## 3.1 PRELIMINARY

### 3.1.1 TASK FORMULATION

Our research addresses the challenge of personalized text generation in resource-constrained local environments. This scenario involves a dual-model architecture where a cloud-based LLM processes only the raw user query, and a locally deployed SLM has access to privacy-sensitive personal context, including user preferences, profiles, and interaction histories.

The primary objective is to optimize generation quality by synergizing the LLM's general linguistic capabilities with localized personalization while maintaining strict privacy preservation. Unlike conventional approaches that frame decoding as a continuous Markov Decision Process (MDP), we reformulate per-token generation as an independent online optimization task. In this paradigm, the LLM's logit distribution during decoding serves as the policy $\pi$, enabling us to formalize the optimization objective for per-token generation as:

$$\pi^*(a) = \arg\max_{\pi \in \Pi} \mathbb{E}_{a \sim \pi(\cdot|p,s)} r\left(a \mid p, s\right) \tag{1}$$

where $p$ denotes the initial prompt, $s$ represents the sequence that has already been generated, $a \in \mathcal{A}$ is a potential token in the vocabulary space $\mathcal{A}$, and $r$ is the latent reward function that reflects the current user's personal context.

### 3.1.2 ONLINE LEARNING

User personalization tasks are characterized by individuality, diversity, and changing over time. Traditional offline learning methods rely on static datasets, which struggle to capture dynamic personalized information in real-time. Online learning, however, enables the model to adapt instantly using the latest personalized context.

Inspired by the recent work Amulet (Zhang et al., 2025b), our study employs the widely studied Follow-The-Regularized-Leader (FTRL) (McMahan, 2011) as the online learning algorithm. The FTRL is the FTL algorithm with a regularizer term added, which significantly improves both algorithmic stability and convergence properties. Typically, the policy optimization process of FTRL at iteration $t$ can be mathematically represented as:

$$\pi_t(a) = \arg\max_{\pi \in \Pi} \left[\sum_{i=0}^{t-1} \mathcal{U}_i(\pi_i(a)) - \frac{1}{\eta}\left(\phi(\pi) + \frac{\lambda}{2}\|\pi - \pi_{t-1}\|^2\right)\right] \tag{2}$$

Given the practical constraint of inaccessible true user rewards, each iteration $t$ necessitates the employment of an approximate utility function $U$ to progressively estimate personalization signals for iterative policy refinement. The first term in Equation 2 implements the fictitious play mechanism to minimize regret between current and historical policies, while the second term represents the regularization and smoothness component with $\eta, \lambda$ as tunable hyperparameters.

## 3.2 DECODING-TIME PERSONALIZATION VIA LOCAL DELTA STEERING

From Equation 1, we understand that the distribution of logits is the strategy we need to optimize. Once we identify an appropriate utility function $U$ to characterize the relative quality of the current logits distribution, we can use Equation 2 to iteratively optimize it.

Series of recent works, including Contrastive Decoding (Li et al., 2023) and Linear Alignment (Gao et al., 2024), have demonstrated that the difference in logits before and after incorporating specific context into the model can serve as a direction for optimization (He et al., 2025; Zhang et al., 2025b). Thus from the online learning perspective, their utility function can be expressed as follows:

$$u_t(a) = \alpha \left(\log \pi_t(a) - \log \pi_{\text{base}}(a)\right) \tag{3}$$

However, this approach is not feasible in our task scenario, as it requires transmitting personal context to the cloud-based LLM, which poses a risk of privacy leakage. Inspired by this idea, we notice that both the small and LLMs share the same state space during decoding. Therefore, we can use delta, the difference in the SLM's logits before and after incorporating personal context, to steer the LLM's logits distribution. Formally, we define the utility function as follows:

$$u_t(a) = \underbrace{\alpha(\log \pi_t(a) - \log \pi_{base}(a))}_{\text{LLM Policy Contrast}} + \underbrace{\beta(\log \pi_{pers}^*(a) - \log \pi_{base}^*(a))}_{\text{SLM Delta Steering}} \qquad (4)$$

Here, we define $p_{base}$ as the user's basic query, and $p_{pers}$ represents the privacy-sensitive personal context. We formalize three distinct generation policies:

- Target Policy $\pi_t(a)$ that is being optimized at step $t$
- LLM Base Policy with query only: $\pi_{base}(a) = P_{LLM}(a|p_{base}, s)$
- SLM Reference Policies:
    - $\pi_{base}^*(a) = P_{SLM}(a|p_{base}, s)$: SLM baseline policy with query only
    - $\pi_{pers}^*(a) = P_{SLM}(a|p_{base}, p_{pers}, s)$: SLM policy with full personal context

where $s$ denotes the current token sequence, $P_{LLM}/\pi$ denotes the LLM's distribution, and $P_{SLM}/\pi^*$ the SLM's distribution. Hyperparameters $\alpha, \beta$ adjust the influence of each optimization component.

The utility function implements a dual-contrastive mechanism: The first term amplifies desirable deviations from the LLM's base policy, while the second term aligns optimization with the personal context-aware SLM policy. Through iterative application, this formulation progressively accentuates the semantic differential induced by $p_{pers}$ relative to $p_{base}$, driving the target policy $\pi_t$ toward maximal utilization of local delta signals.

Additionally, to retain the LLM's stronger general capabilities, we want to ensure that the final strategy does not deviate too far from the LLM's initial strategy. Therefore, we incorporate a KL divergence term into Equation 4 to impose this constraint.

$$\mathcal{U}_t(\pi) = u_t(\pi) - \lambda D_{\text{KL}}(\pi \| \pi_0) \qquad (5)$$

where the initial policy $\pi_0 = \pi_{base}$. The FTRL algorithm requires adding a regularizer to stabilize the algorithm. Here, we continue to use KL divergence as this term, leading to a FTRL-proximal-like iteration dynamic.

$$\pi_t = \arg\max_{\pi \in \Pi} \left[ \sum_{i=0}^{t-1} \mathcal{U}_i(\pi) - \frac{1}{\eta} D_{\text{KL}}(\pi \| \pi_{t-1}) \right] \qquad (6)$$

By substituting Equation 4 into 5, and then 5 into 6, we can iteratively optimize the strategy. However, the current iterative computation incurs significant overhead, which is not feasible for our personalization scenarios that require rapid response. Therefore, we deduce the closed-form solution of Equation 6, as shown in Equation 7, to significantly reduce the additional computational cost. The derivation of the closed-form solution can be found in Appendix A.3.

$$\pi_t(a) \propto \exp\left( \frac{1}{t\lambda + \frac{1}{\eta}} \left( \sum_{i=0}^{t-1} u_i(a) + t\lambda \log \pi_0(a) + \frac{1}{\eta} \log \pi_{t-1}(a) \right) \right) \qquad (7)$$

### 3.3 CoSteer Framework

After explaining how we use local delta signals to steer the target policy, let us formally introduce our **CoSteer** framework, as shown in Algorithm 1. During the inference of each token, the cloud-based LLM sends the current logits to the local environment. The local SLM simultaneously performs inference using prompts with and without personal context to obtain delta signals. Upon receiving the LLM's logits from the cloud, the local environment applies Equation 7 for iterative optimization and samples the output token.

Crucially, this fusion process is performed entirely locally on the edge device. Consequently, the cloud server never receives the raw logit differences or gradients, but only the final, discrete token. This architecture ensures that sensitive personal context remains strictly on-device. Furthermore, since the final token selection happens locally, CoSteer naturally supports the integration of post-processing safeguards (e.g., PII filters) to preemptively inspect the fused token before transmission, ensuring sensitive data remains secure.

Finally, the sampled token is uploaded back to the LLM for inferring the next step. Through this method, CoSteer integrates the large language model's general linguistic capabilities with localized personalization that maintains privacy preservation.

## 4 EXPERIMENT

### 4.1 TASKS AND DATASETS

To demonstrate the versatility of our CoSteer framework, we conduct extensive experiments on **eight** datasets spanning two major categories of personalization tasks: personalized content generation and preference alignment.

**Personalized content generation** We utilize two established benchmarks, Cogenesis (Zhang et al., 2024b) and LongLaMP (Kumar et al., 2024). Cogenesis provides summarized user profiles and past experiences, with the aim of generating highly personalized content. We evaluate on its official test set. LongLaMP focuses on personalized long-form generation, where each writing query includes ground truth responses and the user's previous writing records. We augment each instance with the top-5 relevant historical records retrieved by `bge-reranker-v2-m3` (Chen et al., 2024) as its personal context. We conduct experiments on the official test sets of three constituent tasks: abstract generation (Tang et al., 2008), review writing (Ni et al., 2019), and topic writing (Völske et al., 2017).

**Preference alignment** We evaluate on four datasets: HelpSteer (Wang et al., 2023b), Truthful QA (Lin et al., 2022), UltraChat (Ding et al., 2023), and Personal Preference Eval (Gao et al., 2024). These datasets require generating content aligned with explicitly stated user preferences. Due to the large scale of these datasets, we randomly sample 200 test instances from each dataset.

Detailed dataset description and examples can be found in Appendix B.5

### 4.2 EVALUATION METRICS

To ensure fairness in comparison, we employ task-specific metrics proposed by these datasets themselves for evaluation. For Cogenesis, we use `GPT-4o-2024-08-06` to evaluate the response's overall and personalized scores, averaging the results over five runs with temperature set to 0 to mitigate potential instability. For LongLaMP, we employ ROUGE (Lin, 2004) and METEOR (Banerjee & Lavie, 2005) scores to measure content overlap and semantic alignment, supplementing them with human evaluation (detailed in Appendix B.6) to account for the potential limitations of these automatic metrics. For preference alignment tasks, following prior work (Zhang et al., 2025b; Zhong et al., 2024), we set user preferences to be *concise*, *creative*, *uplifting*, and *verbose*, and utilize the reward model `ArmoRM-8B` (Wang et al., 2024) to evaluate the degree to which the generations align with these preferences.

### 4.3 SETTINGS AND BASELINES

To verify the robustness of our framework, we evaluate five model pairs of varying scales and architectures, including (1) `Qwen2.5-7B-Instruct` with `Qwen2.5-1.5B-Instruct` (2) `Qwen2.5-32B-Instruct` with `Qwen2.5-7B-Instruct` (Team, 2024) (3) `Llama3.1-8B-Instruct` with `Llama-3.2-1B-Instruct` (et al., 2024) (4) `Qwen3-8B` with `Qwen3-0.6B` and (5)`Qwen3-32B` with `Qwen3-0.6B` (Yang et al., 2025). Detailed parameters and settings are presented in Appendix B.1. We establish the following critical baselines per pair:

- SLM w/o: Base performance of standalone SLMs without personal context.

| Models | Setting | Cogenesis | | Abstract Generation | | | Review Writing | | | Topic Writing | | | Pref Align | | | |
|---|---|---|---|---|---|---|---|---|---|---|---|---|---|---|---|---|
| | | Ovl | Per | R-1 | R-L | MET | R-1 | R-L | MET | R-1 | R-L | MET | creative | verbose | concise | uplifting |
| Qwen 7B-1.5B | SLM w/o | 6.63 | 6.21 | 36.48 | 17.74 | 27.57 | 20.40 | 10.39 | 10.39 | 25.21 | 11.09 | 17.51 | .5441 | .5154 | .6104 | .5632 |
| | SLM w/ | 7.81 | 7.63 | 39.75 | 22.03 | 27.76 | 23.08 | 12.40 | 12.94 | 22.89 | 11.46 | 17.12 | .6214 | .6998 | .7102 | .6594 |
| | LLM w/o | 8.00 | 7.63 | 39.81 | 20.53 | 25.56 | 30.15 | 14.04 | 17.71 | 27.64 | 11.93 | 21.49 | .7058 | .6931 | .7039 | .7183 |
| | CoSteer | **8.44** | **8.50** | **42.98** | **23.61** | **28.20** | **32.72** | **15.92** | **20.36** | 25.93 | **12.38** | **22.84** | **.7915** | **.7608** | **.7404** | **.7885** |
| | LLM w/ | 8.62 | 8.60 | 44.50 | 24.63 | 31.15 | 33.83 | 15.55 | 22.42 | 28.82 | 13.77 | 23.44 | .8884 | .8076 | .7693 | .8414 |
| Qwen 32B-7B | SLM w/o | 8.00 | 7.63 | 39.81 | 20.53 | 25.56 | 30.15 | 14.04 | 17.71 | 27.64 | 11.93 | 21.49 | .7058 | .6931 | .7039 | .7183 |
| | SLM w/ | 8.62 | 8.60 | 44.50 | 24.63 | 31.15 | 33.83 | 15.55 | 22.42 | 28.82 | 13.77 | 23.44 | .8884 | .8076 | .7693 | .8414 |
| | LLM w/o | 8.12 | 7.87 | 40.66 | 21.02 | 26.61 | 32.21 | 14.44 | 19.61 | 28.82 | 12.20 | 21.16 | .7020 | .6873 | .7225 | .7146 |
| | CoSteer | **8.78** | **8.64** | **45.41** | **26.04** | **33.52** | **34.88** | **15.89** | **26.51** | 30.10 | **14.52** | **24.20** | .8589 | **.8532** | .7274 | **.8579** |
| | LLM w/ | 8.83 | 8.76 | 43.33 | 23.47 | 30.10 | 34.65 | 15.74 | 22.77 | 30.73 | 14.20 | 24.25 | .9017 | .8193 | .7912 | .8538 |
| Llama 8B-1B | SLM w/o | 7.04 | 6.55 | 33.20 | 18.20 | 28.55 | 31.75 | 14.92 | 19.78 | 20.81 | 10.21 | 17.30 | .6535 | .6355 | .5900 | .6646 |
| | SLM w/ | 7.69 | 7.52 | 39.81 | 21.53 | 30.11 | 32.36 | 15.02 | 22.06 | 20.17 | 10.58 | 18.64 | .7981 | .7908 | .7037 | .8065 |
| | LLM w/o | 7.69 | 7.13 | 39.33 | 20.69 | 29.41 | 34.58 | 15.32 | 22.10 | 26.93 | 12.35 | 21.82 | .7038 | .6878 | .6765 | .7116 |
| | CoSteer | 7.29 | **7.73** | **41.28** | **24.97** | **31.19** | 31.68 | 13.68 | **24.57** | 26.11 | 12.00 | 23.69 | **.8911** | **.8582** | **.7812** | **.8721** |
| | LLM w/ | 8.61 | 8.44 | 43.91 | 23.93 | 32.01 | 36.39 | 15.95 | 23.56 | 30.54 | 14.02 | 23.81 | .8869 | .8298 | .7909 | .8551 |

Table 1: Comparative performance across eight personalized content generation and preference alignment tasks. Metrics include overall (Ovl) and personalized (Per) scores for Cogenesis, ROUGE-1/-L (R-1/-L) and METEOR (MET) for Longlamp datasets, and averaged alignment scores for four user preferences. **Bold** entries indicate that CoSteer outperforms the three baseline methods compared against. Gray values represent the privacy-violating near-upper-bound performance, yet underlined CoSteer values surpasses these incompatible references.

- SLM w/: SLMs augmented with personal context.

- LLM w/o: Cloud-based LLMs without access to personal context.

- LLM w/: The near-upper-bound performance where LLMs directly access personal context. It is important to note that this approach can lead to privacy breaches and does not align with our task setting.

## 4.4 Main results

Table 1 presents our core experimental findings. For clarity, here we present results from first three of our five model pair configurations and report the average performance across the four preference alignment tasks. The complete results, including for the remaining two model pairs (Table 6) and the detailed per-dataset preference alignment scores (Table 5), can be found in the Appendix B.2. Furthermore, we conducted paired t-tests to verify that our improvements are statistically significant, with a detailed analysis available in Appendix B.3.

**Overall Performance** In the vast majority of settings, our method significantly outperforms key baselines: (1) cloud-based LLMs responding without access to personal context, and (2) local SLMs with or without such context. While we observe consistent and robust improvements on standard model pairs (e.g., Qwen2.5 32B-7B and 7B-1.5B), the performance on other configurations exhibits interesting nuances related to model characteristics. For instance, results on the Llama 3 pair vary with output length; it excels in concise *Preference Alignment* tasks but shows more moderate gains in long-form generation, likely due to the compact Llama-1B's stylistic bias towards brevity. Similarly, with the ultra-compact Qwen3-0.6B, while we see strong benefits in complex personalized writing, gains in simpler alignment tasks are less pronounced. We attribute this to CoSteer's regularization mechanism, which conservatively limits steering intensity when the capacity gap is extreme, effectively preventing the tiny SLM from degrading the LLM's general coherence. Remarkably, our performance often approaches or even exceeds the "near-upper-bound" achieved by LLMs with full context access. These results show that CoSteer enables cloud-based LLMs to generate personalized content using

only locally stored user context. We further analyze the impact of task complexity and model scale in Appendix B.4.

### 4.5 COMPARISON WITH ALTERNATIVE METHODS

While CoSteer is **unique in its technical positioning** as shown in Table 3, to comprehensively demonstrate its superiority, we further compare it against other alignment methods, particularly those that rely solely on localized SLMs. Among the various existing techniques, we report on a selection of methods that can be applied to Personalized Content Generation tasks and that outperform the `SLM w/` and `SLM w/o` baselines. These methods include:

1. **Linear Alignment (Gao et al., 2024) and Context Steering(He et al., 2025)** : These two methods are formally equivalent to scaling the logit difference of the SLM (with and without personal context) and applying it to its own inference process. This constitutes an inference-time optimization performed directly on the SLM.

2. **Supervised Fine-Tuning (SFT)**: This involves directly fine-tuning the localized SLMs using personal data. Although we argue that fine-tuning a separate model for each user and task is impractical in real-world scenarios due to privacy and data constraints, we report its performance here for a thorough academic comparison.

3. **Proxy-tuning Liu et al. (2024)**: This method leverages the difference in the SLM's logit distributions before and after the SFT process described above. This difference is directly added to the LLM's logits to steer its output distribution.

We conduct experiments on LongLaMP using the Qwen7B-1.5B configuration. Implementation details and parameter settings for these baselines are provided in the Appendix B.8. As shown in Table 2, the results again confirm the superiority of the CoSteer framework. Our method adeptly combines the performance advantage of localized SLMs in handling private data with the powerful general capabilities of cloud-based LLMs, achieving the best performance among all compared methods without requiring any training.

| | Abstract | | | Review | | | Writing | | |
|---|---|---|---|---|---|---|---|---|---|
| | R-1 | R-L | MET | R-1 | R-L | MET | R-1 | R-L | MET |
| LA/CS | 41.49 | **25.57** | **29.96** | 26.60 | 13.68 | 17.59 | 24.09 | 12.32 | 19.94 |
| SFT | 40.71 | 21.85 | 27.00 | 31.55 | 12.91 | 20.23 | 25.53 | 10.80 | 19.35 |
| Proxy tuning | 40.76 | 21.37 | 25.95 | 29.54 | 14.16 | 16.79 | **27.13** | 12.11 | 21.38 |
| **Costeer** | **42.98** | 23.61 | 28.20 | **32.72** | **15.92** | 20.36 | 25.93 | **12.38** | **22.84** |
| w/o alpha | 41.10 | 21.78 | 26.09 | 30.51 | 14.40 | 17.47 | 27.41 | 12.32 | 21.95 |
| w/o KL | 42.52 | 22.76 | 27.04 | 31.44 | 14.43 | 18.70 | 26.47 | 13.12 | 22.82 |

Table 2: Performance comparison of our Costeer framework against other methods on LongLamp benchmark, with an ablation study on its key components. Among four methods, best results are marked in **bold** while the second-best are underlined.

### 4.6 ABLATION STUDY AND PARAMETER ANALYSIS

**Ablation study**  We conduct an ablation study to validate the contributions of the key components in our framework. The SLM delta steering signal (the $\beta$ term in Eq. (4)) is the central mechanism of CoSteer. Our study therefore focuses on the impact of individually removing two other components: the LLM Alignment Term (the $\alpha$ term) and the KL Divergence Regularizer from the FTRL algorithm. The results in Table 2 proved their effectiveness. A broader ablation on the entire iterative FTRL (we name it LightCosteer) is discussed in Section 5.3.

**Hyperparameters**  We examine the impact of different hyperparameters on our method. We conduct experiments on abstract generation using the Qwen7B-1.5B configuration, and evaluate with the ROUGE-L metric. When studying one parameter, we keep all others at their default values. Results and analysis presented in Appendix B.9 and Figure 2 show consistently strong performance across a broad range of values, confirming that the default hyperparameter settings (provided in Appendix B.1) are both robust and well-calibrated.

## 5 DISCUSSION: PRACTICAL DEPLOYMENT CONSIDERATIONS

In Section 4, we showed the empirical superiority of the CoSteer framework through comprehensive experiments and comparisons. In this section, we shift our focus to its practical viability, analyzing its feasibility and performance from various angles that simulate real-world conditions and constraints.

### 5.1 ROBUSTNESS TO NOISY USER CONTEXT

In practical on-device applications, the user-specific context is often imperfect and susceptible to noise. To evaluate **CoSteer**'s stability in such scenarios, we conducted rigorous stress tests under two distinct noise conditions: (1) **realistic noise**, simulated by replacing the strong `bge-m3` retriever with a weaker `BM25` retriever, and (2) **adversarial noise**, where we intentionally provided completely irrelevant context from a disjoint task.

Our experiments, detailed in Appendix B.10, yield a crucial insight: **CoSteer** demonstrates remarkable resilience to noisy context. Under realistic noise, it consistently outperforms baselines and maintains its performance edge (Table 10). In the more extreme adversarial setting, the framework degrades gracefully rather than collapsing, indicating that LLM guidance prevents the framework from being completely misled by faulty context (Table 11). This robustness is vital for real-world deployment.

### 5.2 GENERALIZATION ACROSS MODEL SCALES AND ARCHITECTURES

A key advantage of our framework is its ability to generalize across diverse model configurations, a critical feature for real-world deployment. We validate this generalization capability in two dimensions: model scale and architecture.

**Generalization Across Model Scales**  Our primary experiments (see Table 6) already demonstrate remarkable scale generalization by pairing a powerful 32B-parameter LLM with a highly compact 0.6B SLM (Qwen3-0.6B), one of the smallest state-of-the-art models available. Despite being over 50× smaller, this SLM effectively steers the LLM across all our testbeds, strongly substantiates the framework's robustness to vast differences in model scale.

**Collaboration Across Model Architectures**  While pairing models from the same family is common, a practical framework should accommodate scenarios where the LLM and SLM differ in architectures and tokenizers. To this end, we investigated two strategies. The first, `CoSteer_map`, aligns models via vocabulary intersection, while effective, its applicability is limited when tokenizers are highly divergent. Therefore, we propose `CoSteer_byte` which operates on a shared byte representation, making it vocabulary-agnostic (Hayase et al., 2025). As shown in Table 12, both strategies successfully enable the SLM to guide the LLM, confirming that **CoSteer is not confined to homogeneous model families**. This flexibility demonstrates the framework's generalization across architectures. Full implementation details are provided in Appendix B.11.

### 5.3 EFFICIENCY AND PRACTICAL VARIANTS

Finally, we analyze the efficiency and overhead of our framework. At a high level, a key advantage of CoSteer is its computational efficiency. Compared to methods that require concatenating long personal context to the LLM's input (i.e., `LLM w/`), our framework offers significant computational savings. By processing the private context locally with an SLM, **CoSteer substantially reduces the remote LLM's workload**. Our FLOPs estimation (Narayanan et al., 2021) in Table 15 shows that with a 1000-token context, the privacy-violating `LLM w/` approach is approximately **3.3x** more computationally expensive than CoSteer.

Although our closed-form solution is efficient, the full CoSteer framework still employs iterative optimization (e.g., $T = 20$), which can introduce latency. Furthermore, each generation step requires communication between the local device and the server. To address these practical overheads, we propose and evaluate two streamlined variants:

- **LightCoSteer**: To minimize computational overhead, this variant eliminates the iterative process by setting the number of optimization iterations $T = 1$. This simplifies the framework to a single-step logit adjustment, providing a lightweight yet effective alternative.

- **AdaCoSteer**: To reduce communication overhead, this variant adaptively deactivates steering. We observe that as generation proceeds, SLM and LLM predictions often converge. AdaCoSteer leverages this by terminating the steering process once the LLM's token confidence exceeds a set threshold for a few consecutive steps (Song et al., 2025), allowing the LLM to complete the generation on its own.

The implementation details and a full analysis of the results are presented in Appendix B.12. Our findings show that these variants create a valuable performance-efficiency spectrum, allowing users to select the optimal configuration that balances effectiveness and resource constraints for diverse application requirements.

## 6 CONCLUSION

In this work, we introduced CoSteer, a collaborative framework that enables real-time, privacy-preserving personalization for LLMs via localized delta steering. Our approach leverages an on-device SLM to guide a cloud-based LLM, effectively incorporating user context without transmitting sensitive data or requiring fine-tuning. Extensive experiments demonstrate that CoSteer offers a practical and effective solution for harmonizing the powerful general capabilities of LLMs with user-specific context and privacy.

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

APPENDICES CONTENTS

# A  METHODOLOGY DETAIL

## A.1  RELATED WORK AND DISTINCTION

As summarized in Table 3, **CoSteer** occupies a unique technical position by being both collaborative and completely tuning-free. While it shares conceptual roots with inference-time steering methods, it introduces critical innovations tailored to the privacy-constrained cloud-edge scenario.

**Distinction from Context Steering Methods**   Unlike methods like Context Steering (He et al., 2025) or Linear Alignment (Gao et al., 2024), which operate in a single-model setting where a model steers itself, CoSteer addresses a fundamentally different challenge: **heterogeneous cross-model collaboration**. We demonstrate a novel "weak-to-strong" capability where a tiny, local SLM effectively steers a massive, remote LLM. This allows high-quality personalization without exposing private data to the cloud, a constraint that single-model cloud deployment cannot satisfy.

**Technical Novelty beyond Optimization**   While we adopt the FTRL algorithm as our solver, our core contribution lies in the **problem formulation**. We novelly define the utility function (Equation 4) based on local logit deltas to repurpose online learning for solving a distributed coordination problem. Furthermore, to address practical bottlenecks unique to this collaborative architecture, we introduce system-level innovations not found in prior work: **AdaCoSteer** minimizes communication latency via confidence-based termination, and **Byte-Level Fusion** resolves tokenizer mismatches to enable collaboration between heterogeneous model families (e.g., Llama and Qwen).

Table 3: Our Technical position. The main body of the table uses "Yes/No" for clarity. Explanations for specific "No" entries are as follows: [a]Requires a trained reward model. [b]Requires a fine-tuned smaller language model (SLM). [c]Requires a trained fusion network.

| Method | Training Free? | Weak-to-Strong Collaborative? | No Additional Modules? |
|---|---|---|---|
| Linear Alignment (Gao et al., 2024) | | | |
| Contrastive Decoding (Li et al., 2023) | Yes | No | Yes |
| Context Steering (He et al., 2025) | | | |
| Amulet (Zhang et al., 2025b) | | | |
| PAD (Chen et al., 2025) | No[a] | No | No |
| Drift (Kim et al., 2025) | | | |
| Proxy-tuning (Liu et al., 2024) | No[b] | Yes | Yes |
| Cogenesis (Zhang et al., 2024b) | No[c] | Yes | No |
| **CoSteer (Ours)** | **Yes** | **Yes** | **Yes** |

## A.2 FRAMEWORK

Our CoSteer framework workflow is shown in Algorithm 1

---
**Algorithm 1** CoSteer Framework
---
**Require:**
  Cloud-based LLM policy generator $\pi_{\text{LLM}}$, local SLM policy generator $\pi_{\text{SLM}}$, user query $p_{\text{base}}$,
  personal context $p_{\text{pers}}$, current sequence $s$, max tokens $M$, hyperparameters: T, $\alpha, \beta, \lambda, \eta > 0$
**Ensure:** Personalized sequence $s$
 1: Initialize $s \leftarrow \emptyset$
 2: **repeat**
 3:  $\pi_{\text{base}} \leftarrow \pi_{\text{LLM}}(a|p_{\text{base}}, s)$                  $\triangleright$ Cloud computation
 4:  $\pi_{\text{base}}^* \leftarrow \pi_{\text{SLM}}(a|p_{\text{base}}, s)$                  $\triangleright$ Edge computation
 5:  $\pi_{\text{pers}}^* \leftarrow \pi_{\text{SLM}}(a|p_{\text{base}}, p_{\text{pers}}, s)$
 6:  $\Delta \leftarrow \log \pi_{\text{pers}}^* - \log \pi_{\text{base}}^*$
 7:  $\pi_0 \leftarrow \pi_{\text{base}}$                       $\triangleright$ Initialize policy
 8:  **for** $t = 1$ **to** $T$ **do**
 9:    $u_t \leftarrow \alpha(\log \pi_{t-1} - \log \pi_{\text{base}}) + \beta\Delta$
10:    Update policy using Equation 7
11:  **end for**
12:  $\pi^* \leftarrow \pi_T$                      $\triangleright$ Final optimized policy
13:  Sample $a \sim \pi^*$, Update $s \leftarrow s \circ a$
14: **until** len$(s) \geq M$ **or** EOS generated
15: **return** $s$

---

## A.3 PROOF

Similar to Zhang et al. (2025b), We try to solve the closed-form solution of Equation 6:

$$\mathcal{L}(\pi_t, \mu) = \underbrace{\sum_{i=0}^{t-1}\sum_{a \in A} \pi_t(a)u_i(a)}_{(1)} - t\lambda \underbrace{\sum_{a \in A}\left(\pi_t(a)\log\frac{\pi_t(a)}{\pi_0(a)}\right)}_{(2)}$$
$$- \underbrace{\frac{1}{\eta}\sum_{a \in A}\pi_t(a)\log\frac{\pi_t(a)}{\pi_{t-1}(a)}}_{(3)} + \underbrace{\mu\left(1 - \sum_{a \in A}\pi_i(a)\right)}_{(4)} \tag{8}$$

Here, (1) and (2) originate from the utility function $\mathcal{U}$, (3) from the KL divergence, and (4) constrains the sum to 1, with the Lagrange multiplier $\mu$. We calculate the derivation of the function for a given $a$, we have

$$\frac{\partial\mathcal{L}(\pi_t, \mu)}{\partial\pi_t(a)} = \sum_{i=0}^{t-1} u_i(a) - t\lambda\left(\log\frac{\pi_t(a)}{\pi_0(a)} + 1\right) - \frac{1}{\eta}\left(\log\frac{\pi_t(a)}{\pi_{t-1}(a)} + 1\right) - \mu \tag{9}$$

Rearrange the terms:

$$\sum_{i=0}^{t-1} u_i(a) - t\lambda\log\pi_t(a) + t\lambda\log\pi_0(a) - \frac{1}{\eta}\log\pi_t(a) + \frac{1}{\eta}\log\pi_{t-1}(a) - t\lambda - \frac{1}{\eta} - \mu = 0 \tag{10}$$

Combine the coefficients of $\log\pi_t(a)$ :

$$-\left(t\lambda + \frac{1}{\eta}\right)\log\pi_t(a) = -\sum_{i=0}^{t-1} u_i(a) - t\lambda\log\pi_0(a) - \frac{1}{\eta}\log\pi_{t-1}(a) + t\lambda + \frac{1}{\eta} + \mu \tag{11}$$

Solve for $\log\pi_t(a)$ :

$$\log \pi_t(a) = \frac{1}{t\lambda + \frac{1}{\eta}} \left( \sum_{i=0}^{t-1} u_i(a) + t\lambda \log \pi_0(a) + \frac{1}{\eta} \log \pi_{t-1}(a) - t\lambda - \frac{1}{\eta} - \mu \right) \quad (12)$$

let

$$C = -\frac{t\lambda + \frac{1}{\eta} + \mu}{t\lambda + \frac{1}{\eta}} \quad (13)$$

we have

$$\log \pi_t(a) = \frac{1}{t\lambda + \frac{1}{\eta}} \left( \sum_{i=0}^{t-1} u_i(a) + t\lambda \log \pi_0(a) + \frac{1}{\eta} \log \pi_{t-1}(a) \right) + C \quad (14)$$

Thus

$$\pi_t(a) \propto exp \left( \frac{1}{t\lambda + \frac{1}{\eta}} \left( \sum_{i=0}^{t-1} u_i(a) + t\lambda \log \pi_0(a) + \frac{1}{\eta} \log \pi_{t-1}(a) \right) \right) \quad (15)$$

Specifically, let $T = 1$ and $\beta^* = \frac{\beta}{\lambda + \frac{1}{\eta}}$, the policy is simplified to

$$\pi(a) \propto exp \left( \log \pi_0(a) + \beta^* \left( \log \pi_{pers}^*(a) - \log \pi_{base}^*(a) \right) \right) \quad (16)$$

## B  EXPERIMENT DETAIL

### B.1  HYPERPARAMETERS AND SETTINGS

To ensure the results are easily reproducible, we set `do_sample=False` for all methods. For the Qwen3 series, we set `enable_thinking=False`. Hyperparameters of CoSteer are listed in Table 4

Table 4: Default hyperparameter of CoSteer

| Parameter | Value |
|-----------|-------|
| T | 20 |
| $\alpha$ | 2.0 |
| $\beta$ | 1.0 |
| $\lambda$ | 2 |
| $\eta$ | 10 |

### B.2  DETAILED RESULTS

Detailed results on HelpSteer, Personal_Eval, Truthful QA and Ultrachat are shown in Table 5.

| Models | Setting | HelpSteer | | | | Personal_Eval | | | | Truthful QA | | | | Ultrachat | | | |
|---|---|---|---|---|---|---|---|---|---|---|---|---|---|---|---|---|---|
| | | creative | verbose | concise | uplifting | creative | verbose | concise | uplifting | creative | verbose | concise | uplifting | creative | verbose | concise | uplifting |
| Qwen 7B-1.5B | SLM w/o | .5609 | .5301 | .6437 | .5638 | .6043 | .5882 | .6556 | .6722 | .4468 | .4231 | .5349 | .4544 | .5642 | .5202 | .6074 | .5625 |
| | SLM w/ | .6309 | .7093 | .7110 | .6568 | .6564 | .7402 | .7686 | .7449 | .5419 | .6515 | .6358 | .5758 | .6562 | .6982 | .7252 | .6602 |
| | LLM w/o | .7368 | .7223 | .7228 | .7338 | .7550 | .7462 | .7005 | .8124 | .5739 | .5752 | .6744 | .5782 | .7576 | .7285 | .7179 | .7489 |
| | CoSteer | **.8093** | **.7787** | **.7731** | **.8055** | **.8472** | **.7945** | **.7525** | **.8739** | **.6834** | **.6930** | **.6837** | **.6682** | **.8259** | **.7770** | **.7522** | **.8064** |
| | LLM w/ | .8918 | .8173 | .7853 | .8516 | .9217 | .8396 | .8001 | .9073 | .8363 | .7593 | .7069 | .7630 | .9037 | .8140 | .7850 | .8438 |
| Qwen 32B-7B | SLM w/o | .7368 | .7223 | .7228 | .7338 | .7550 | .7462 | .7005 | .8124 | .5739 | .5752 | .6744 | .5782 | .7576 | .7285 | .7179 | .7489 |
| | SLM w/ | .8918 | .8173 | .7853 | .8516 | .9217 | .8396 | .8001 | .9073 | .8363 | .7593 | .7069 | .7630 | .9037 | .8140 | .7850 | .8438 |
| | LLM w/o | .7141 | .7003 | .7406 | .7129 | .7548 | .7461 | .7129 | .8136 | .5877 | .5844 | .6993 | .5925 | .7512 | .7183 | .7373 | .7393 |
| | CoSteer | .8630 | **.8516** | .7478 | **.8684** | .9158 | **.8754** | .7700 | **.9166** | .7985 | **.8425** | .6575 | .7818 | .8581 | **.8433** | .7344 | **.8646** |
| | LLM w/ | .9038 | .8270 | .8104 | .8628 | .9254 | .8434 | .8215 | .9117 | .8698 | .7895 | .7208 | .7897 | .9078 | .8171 | .8122 | .8510 |
| Llama 8B-1B | SLM w/o | .6779 | .6595 | .6063 | .6764 | .7297 | .7159 | .6378 | .7915 | .5215 | .5084 | .5119 | .5064 | .6850 | .6581 | .6039 | .6842 |
| | SLM w/ | .8070 | .7915 | .7161 | .8083 | .8604 | .8529 | .7805 | .8866 | .7115 | .7291 | .6023 | .7263 | .8134 | .7897 | .7159 | .8049 |
| | LLM w/o | .7227 | .7110 | .6895 | .7169 | .7476 | .7355 | .6689 | .8021 | .6044 | .5909 | .6574 | .5967 | .7405 | .7137 | .6903 | .7306 |
| | CoSteer | **.8968** | **.8571** | .7993 | **.8735** | **.9173** | **.8788** | **.8297** | **.9235** | **.8499** | **.8408** | .6951 | **.8266** | **.9004** | **.8562** | **.8008** | **.8648** |
| | LLM w/ | .8951 | .8375 | .8121 | .8603 | .9030 | .8556 | .8108 | .9004 | .8634 | .7974 | .7410 | .8101 | .8860 | .8285 | .7997 | .8496 |
| Qwen 0.6B-8B | SLM w/o | .5844 | .5477 | .5984 | .5945 | .6498 | .6291 | .6573 | .7194 | .4014 | .3648 | .5039 | .4144 | .6485 | .6018 | .6327 | .6463 |
| | SLM w/ | .6255 | .6428 | .6099 | .6313 | .7299 | .7302 | .6614 | .7638 | .4928 | .5277 | .4942 | .4968 | .7063 | .6721 | .6343 | .6826 |
| | LLM w/o | .7819 | .7783 | .7084 | .7771 | .8057 | .8037 | .6920 | .8448 | .6536 | .6504 | .6675 | .6461 | .7974 | .7796 | .7038 | .7765 |
| | CoSteer | .7849 | **.7806** | .7046 | .7757 | **.8058** | **.8117** | .6901 | **.8477** | **.6582** | **.6549** | .6647 | **.6506** | **.7986** | **.7819** | .7058 | **.7853** |
| | LLM w/ | .8928 | .8037 | .7842 | .8629 | .9076 | .8303 | .8141 | .9020 | .8612 | .7796 | .6952 | .8004 | .8964 | .7989 | .7915 | .8555 |
| Qwen 0.6B-32B | SLM w/o | .5844 | .5477 | .5984 | .5945 | .6498 | .6291 | .6573 | .7194 | .4014 | .3648 | .5039 | .4144 | .6485 | .6018 | .6327 | .6463 |
| | SLM w/ | .6255 | .6428 | .6099 | .6313 | .7299 | .7302 | .6614 | .7638 | .4928 | .5277 | .4942 | .4968 | .7063 | .6721 | .6343 | .6826 |
| | LLM w/o | .8103 | .8084 | .6991 | .7922 | .8179 | .8237 | .6948 | .8443 | .7084 | .7130 | .6863 | .7010 | .8077 | .7948 | .6935 | .7758 |
| | CoSteer | .8083 | **.8152** | .6984 | **.7923** | .8145 | .8191 | .6752 | .8436 | **.7152** | **.7226** | **.6870** | **.7020** | .8053 | **.7962** | .6884 | **.7779** |
| | LLM w/ | .9121 | .8147 | .8294 | .8722 | .9094 | .8352 | .8298 | .8936 | .9097 | .7886 | .7364 | .8327 | .9069 | .7996 | .8204 | .8629 |

Table 5: Results on all four preference alignment datasets with our proposed CoSteer and other settings. **Bold** values indicate that our method outperformed the three baseline methods we are comparing against. Note that results from the large model with context are marked in gray, as this scenario does not align with our task setting.

| Models | Setting | Cogenesis | | Abstract Generation | | | Review Writing | | | Topic Writing | | | Pref Align | | | |
|---|---|---|---|---|---|---|---|---|---|---|---|---|---|---|---|---|
| | | Ovl | Per | R-1 | R-L | MET | R-1 | R-L | MET | R-1 | R-L | MET | creative | verbose | concise | uplifting |
| Qwen 8B-0.6B | SLM w/o | 6.84 | 6.64 | 37.13 | 20.41 | 21.00 | 24.41 | 12.83 | 12.39 | 24.19 | 11.88 | 15.74 | .5710 | .5359 | .5981 | .5937 |
| | SLM w/ | 7.85 | 7.79 | 42.48 | 23.25 | 26.58 | 25.04 | 13.36 | 13.02 | 26.17 | 13.46 | 18.40 | .6386 | .6432 | .6000 | .6436 |
| | LLM w/o | 8.27 | 7.99 | 40.76 | 21.23 | 25.77 | 30.89 | 14.47 | 16.98 | 28.47 | 12.85 | 21.49 | .7597 | .7530 | .6929 | .7611 |
| | CoSteer | **8.62** | **8.65** | 41.11 | 22.36 | 25.86 | **32.24** | **14.84** | **18.47** | **29.03** | **13.61** | **22.98** | **.7619** | **.7573** | .6913 | **.7648** |
| | LLM w/ | 8.96 | 8.96 | 43.99 | 23.69 | 29.48 | 35.42 | 15.71 | 21.59 | 31.48 | 15.34 | 24.78 | .8895 | .8031 | .7713 | .8552 |
| Qwen 32B-0.6B | SLM w/o | 6.84 | 6.64 | 37.13 | 20.41 | 21.00 | 24.41 | 12.83 | 12.39 | 24.19 | 11.88 | 15.74 | .5710 | .5359 | .5981 | .5937 |
| | SLM w/ | 7.85 | 7.79 | 42.48 | 23.25 | 26.58 | 25.04 | 13.36 | 13.02 | 26.17 | 13.46 | 18.40 | .6386 | .6432 | .6000 | .6436 |
| | LLM w/o | 8.31 | 8.22 | 41.05 | 21.88 | 26.23 | 30.23 | 14.08 | 16.41 | 29.01 | 12.51 | 21.44 | .7861 | .7850 | .6934 | .7783 |
| | CoSteer | **8.52** | **8.56** | **43.74** | **23.69** | 30.97 | 30.89 | **14.14** | **18.87** | 26.75 | 13.17 | 21.32 | .7858 | **.7883** | .6873 | **.7790** |
| | LLM w/ | 9.09 | 9.03 | 44.46 | 23.95 | 30.15 | 35.12 | 15.60 | 20.65 | 32.63 | 15.37 | 25.07 | .9095 | .8095 | .8040 | .8654 |

Table 6: Comparative performance across eight personalized content generation and preference alignment tasks. Metrics include overall (Ovl) and personalized (Per) scores for Cogenesis, ROUGE-1/-L (R-1/-L) and METEOR (MET) for LongLaMP datasets, and averaged alignment scores for four user preferences. **Bold** entries indicate that CoSteer outperforms the three baseline methods compared against. Gray values represent the privacy-violating near-upper-bound performance, yet underlined CoSteer values surpasses these incompatible references.

| Models | Setting | Cogenesis | | Abstract Generation | | | Review Writing | | | Topic Writing | | | Pref Align | | | |
|---|---|---|---|---|---|---|---|---|---|---|---|---|---|---|---|---|
| | | Ovl | Per | R-1 | R-L | MET | R-1 | R-L | MET | R-1 | R-L | MET | creative | verbose | concise | uplifting |
| Qwen 7B-1.5B | SLM w/o | $6.63_{\pm1.70}$ | $6.21_{\pm1.88}$ | $36.48_{\pm7.50}$ | $17.74_{\pm3.85}$ | $27.57_{\pm5.44}$ | $20.40_{\pm8.88}$ | $10.39_{\pm3.71}$ | $10.39_{\pm6.07}$ | $25.21_{\pm9.55}$ | $11.09_{\pm3.24}$ | $17.51_{\pm8.06}$ | $.5441_{\pm.13}$ | $.5154_{\pm.11}$ | $.6104_{\pm.09}$ | $.5632_{\pm.06}$ |
| | SLM w/ | $7.81_{\pm1.24}$ | $7.63_{\pm1.43}$ | $39.75_{\pm9.83}$ | $22.03_{\pm9.83}$ | $27.76_{\pm10.03}$ | $23.08_{\pm9.95}$ | $12.40_{\pm7.15}$ | $12.94_{\pm8.90}$ | $22.89_{\pm9.19}$ | $11.46_{\pm6.30}$ | $17.12_{\pm8.45}$ | $.6214_{\pm.10}$ | $.6998_{\pm.08}$ | $.7102_{\pm.13}$ | $.6594_{\pm.05}$ |
| | LLM w/o | $8.00_{\pm1.18}$ | $7.63_{\pm1.43}$ | $39.81_{\pm6.86}$ | $20.53_{\pm4.32}$ | $25.56_{\pm7.11}$ | $30.15_{\pm6.64}$ | $14.04_{\pm2.86}$ | $17.71_{\pm6.54}$ | $27.64_{\pm8.70}$ | $11.93_{\pm2.48}$ | $21.49_{\pm5.51}$ | $.7058_{\pm.10}$ | $.6931_{\pm.08}$ | $.7039_{\pm.13}$ | $.7183_{\pm.07}$ |
| | CoSteer | $\mathbf{8.44^*}_{\pm1.34}$ | $\mathbf{8.50^*}_{\pm1.20}$ | $\mathbf{42.98^*}_{\pm8.94}$ | $\mathbf{23.61^*}_{\pm7.09}$ | $\mathbf{28.20^*}_{\pm8.88}$ | $\mathbf{32.72^*}_{\pm9.67}$ | $\mathbf{15.92^*}_{\pm9.02}$ | $\mathbf{20.36^*}_{\pm10.09}$ | $25.93_{\pm10.13}$ | $\mathbf{12.38^*}_{\pm9.78}$ | $\mathbf{22.84^*}_{\pm10.41}$ | $\mathbf{.7915^*}_{\pm.06}$ | $\mathbf{.7608^*}_{\pm.05}$ | $\mathbf{.7404^*}_{\pm.06}$ | $\mathbf{.7885^*}_{\pm.14}$ |
| | LLM w/ | $8.62_{\pm0.63}$ | $8.61_{\pm0.58}$ | $44.50_{\pm0.77}$ | $24.63_{\pm9.14}$ | $31.15_{\pm9.34}$ | $33.83_{\pm5.44}$ | $15.55_{\pm2.41}$ | $22.42_{\pm4.76}$ | $28.82_{\pm11.76}$ | $13.77_{\pm9.37}$ | $23.44_{\pm8.29}$ | $.8884_{\pm.07}$ | $.8076_{\pm.05}$ | $.7693_{\pm.07}$ | $.8414_{\pm.16}$ |
| Qwen 32B-7B | SLM w/o | $8.00_{\pm1.18}$ | $7.63_{\pm1.43}$ | $39.81_{\pm6.86}$ | $20.53_{\pm4.32}$ | $25.56_{\pm7.11}$ | $30.15_{\pm6.64}$ | $14.04_{\pm2.86}$ | $17.71_{\pm6.54}$ | $27.64_{\pm8.70}$ | $11.93_{\pm2.48}$ | $21.49_{\pm5.51}$ | $.7058_{\pm.09}$ | $.6931_{\pm.09}$ | $.7039_{\pm.08}$ | $.7183_{\pm.15}$ |
| | SLM w/ | $8.62_{\pm0.63}$ | $8.61_{\pm0.58}$ | $44.50_{\pm0.77}$ | $24.63_{\pm9.14}$ | $31.15_{\pm9.34}$ | $33.83_{\pm5.44}$ | $15.55_{\pm2.41}$ | $22.42_{\pm4.76}$ | $28.82_{\pm11.76}$ | $13.77_{\pm9.37}$ | $23.44_{\pm8.29}$ | $.8884_{\pm.05}$ | $.8076_{\pm.06}$ | $.7693_{\pm.12}$ | $.8414_{\pm.05}$ |
| | LLM w/o | $8.12_{\pm1.01}$ | $7.87_{\pm1.16}$ | $40.66_{\pm6.79}$ | $21.02_{\pm4.26}$ | $26.61_{\pm6.83}$ | $32.21_{\pm6.32}$ | $14.44_{\pm2.51}$ | $19.61_{\pm6.44}$ | $28.82_{\pm8.53}$ | $12.20_{\pm2.68}$ | $21.16_{\pm6.11}$ | $.7020_{\pm.14}$ | $.6873_{\pm.06}$ | $.7225_{\pm.10}$ | $.7146_{\pm.13}$ |
| | CoSteer | $\mathbf{8.78^*}_{\pm0.54}$ | $\mathbf{8.64^*}_{\pm0.79}$ | $\underline{\mathbf{45.41^*}}_{\pm9.75}$ | $\mathbf{26.04^*}_{\pm9.99}$ | $\mathbf{33.52^*}_{\pm9.64}$ | $\mathbf{34.88^*}_{\pm6.80}$ | $\mathbf{15.89^*}_{\pm4.72}$ | $\mathbf{26.51^*}_{\pm7.23}$ | $\mathbf{30.10^*}_{\pm12.05}$ | $\mathbf{14.52^*}_{\pm10.76}$ | $\mathbf{24.20^*}_{\pm11.26}$ | $.8589_{\pm.08}$ | $\mathbf{.8532^*}_{\pm.08}$ | $.7274_{\pm.06}$ | $\mathbf{.8579^*}_{\pm.04}$ |
| | LLM w/ | $8.83_{\pm0.48}$ | $8.76_{\pm0.51}$ | $43.33_{\pm9.16}$ | $23.47_{\pm6.40}$ | $30.10_{\pm7.36}$ | $34.65_{\pm5.63}$ | $15.74_{\pm3.83}$ | $22.77_{\pm5.48}$ | $30.73_{\pm10.63}$ | $14.20_{\pm8.45}$ | $24.25_{\pm8.67}$ | $.9017_{\pm.03}$ | $.8193_{\pm.07}$ | $.7912_{\pm.12}$ | $.8538_{\pm.06}$ |
| Llama 8B-1B | SLM w/o | $7.04_{\pm1.65}$ | $6.55_{\pm1.81}$ | $33.20_{\pm8.60}$ | $18.20_{\pm3.78}$ | $28.55_{\pm4.84}$ | $31.75_{\pm5.95}$ | $14.92_{\pm2.37}$ | $22.06_{\pm5.53}$ | $20.81_{\pm10.10}$ | $10.21_{\pm3.22}$ | $17.30_{\pm8.09}$ | $.6535_{\pm.12}$ | $.6355_{\pm.11}$ | $.5900_{\pm.10}$ | $.6646_{\pm.12}$ |
| | SLM w/ | $7.69_{\pm1.40}$ | $7.52_{\pm1.39}$ | $39.81_{\pm9.63}$ | $21.53_{\pm4.95}$ | $30.11_{\pm6.45}$ | $32.36_{\pm6.59}$ | $15.02_{\pm2.70}$ | $22.06_{\pm4.80}$ | $20.17_{\pm10.43}$ | $10.58_{\pm5.99}$ | $18.64_{\pm8.61}$ | $.7981_{\pm.11}$ | $.7908_{\pm.08}$ | $.7037_{\pm.13}$ | $.8065_{\pm.10}$ |
| | LLM w/o | $7.69_{\pm1.38}$ | $7.13_{\pm1.62}$ | $39.33_{\pm8.11}$ | $20.69_{\pm3.90}$ | $29.41_{\pm5.32}$ | $34.58_{\pm5.19}$ | $15.32_{\pm2.43}$ | $22.10_{\pm5.54}$ | $26.93_{\pm10.11}$ | $12.35_{\pm3.40}$ | $21.82_{\pm6.47}$ | $.7038_{\pm.09}$ | $.6878_{\pm.06}$ | $.6765_{\pm.09}$ | $.7116_{\pm.09}$ |
| | CoSteer | $7.29_{\pm1.42}$ | $\mathbf{7.73^*}_{\pm1.41}$ | $\mathbf{41.28^*}_{\pm8.36}$ | $\mathbf{22.97^*}_{\pm8.10}$ | $\mathbf{31.19^*}_{\pm6.42}$ | $31.68_{\pm6.33}$ | $13.68_{\pm4.63}$ | $\mathbf{24.57^*}_{\pm6.93}$ | $26.11_{\pm11.18}$ | $12.00_{\pm6.74}$ | $23.81_{\pm7.39}$ | $\mathbf{.8911^*}_{\pm.05}$ | $\mathbf{.8582^*}_{\pm.08}$ | $\mathbf{.7812^*}_{\pm.12}$ | $\mathbf{.8721^*}_{\pm.10}$ |
| | LLM w/ | $8.61_{\pm0.66}$ | $8.44_{\pm0.80}$ | $43.91_{\pm6.09}$ | $23.93_{\pm6.42}$ | $32.01_{\pm7.97}$ | $36.39_{\pm5.16}$ | $15.95_{\pm2.50}$ | $23.56_{\pm6.40}$ | $30.54_{\pm10.43}$ | $14.02_{\pm6.15}$ | $23.81_{\pm7.30}$ | $.8869_{\pm.06}$ | $.8298_{\pm.05}$ | $.7909_{\pm.07}$ | $.8551_{\pm.06}$ |
| Qwen 8B-0.6B | SLM w/o | $6.84_{\pm1.70}$ | $6.64_{\pm1.80}$ | $37.13_{\pm7.17}$ | $20.41_{\pm4.90}$ | $21.00_{\pm6.91}$ | $24.41_{\pm7.88}$ | $12.83_{\pm3.69}$ | $12.39_{\pm5.68}$ | $24.19_{\pm9.46}$ | $11.88_{\pm3.88}$ | $15.74_{\pm6.38}$ | $.5710_{\pm.13}$ | $.5359_{\pm.04}$ | $.5981_{\pm.10}$ | $.5937_{\pm.11}$ |
| | SLM w/ | $7.85_{\pm1.49}$ | $7.79_{\pm1.42}$ | $42.48_{\pm8.39}$ | $23.25_{\pm6.21}$ | $26.58_{\pm8.29}$ | $25.04_{\pm10.20}$ | $13.36_{\pm7.38}$ | $13.02_{\pm8.71}$ | $26.17_{\pm11.92}$ | $13.46_{\pm9.01}$ | $18.40_{\pm10.13}$ | $.6386_{\pm.12}$ | $.6432_{\pm.14}$ | $.6000_{\pm.14}$ | $.6436_{\pm.12}$ |
| | LLM w/o | $8.27_{\pm1.05}$ | $7.99_{\pm0.95}$ | $40.76_{\pm7.06}$ | $21.23_{\pm4.37}$ | $25.77_{\pm6.75}$ | $30.89_{\pm6.69}$ | $14.47_{\pm2.85}$ | $16.98_{\pm5.79}$ | $28.47_{\pm9.30}$ | $12.85_{\pm2.94}$ | $21.49_{\pm6.05}$ | $.7597_{\pm.15}$ | $.7530_{\pm.14}$ | $.6929_{\pm.15}$ | $.7611_{\pm.07}$ |
| | CoSteer | $\mathbf{8.62^*}_{\pm0.50}$ | $\mathbf{8.65^*}_{\pm1.12}$ | $41.11_{\pm6.51}$ | $22.36_{\pm6.21}$ | $25.86_{\pm7.98}$ | $\mathbf{32.24^*}_{\pm7.22}$ | $\mathbf{14.84^*}_{\pm3.03}$ | $\mathbf{18.47^*}_{\pm6.15}$ | $\mathbf{29.03^*}_{\pm11.07}$ | $\mathbf{13.61^*}_{\pm7.06}$ | $\mathbf{22.98^*}_{\pm8.47}$ | $\mathbf{.7619^*}_{\pm.06}$ | $\mathbf{.7573^*}_{\pm.06}$ | $.6913_{\pm.06}$ | $\mathbf{.7648^*}_{\pm.11}$ |
| | LLM w/ | $8.96_{\pm0.46}$ | $8.96_{\pm0.36}$ | $43.99_{\pm8.46}$ | $23.69_{\pm6.42}$ | $29.48_{\pm7.98}$ | $35.42_{\pm5.45}$ | $15.71_{\pm2.55}$ | $21.59_{\pm4.96}$ | $31.48_{\pm13.00}$ | $15.34_{\pm11.50}$ | $24.78_{\pm11.46}$ | $.8895_{\pm.06}$ | $.8031_{\pm.12}$ | $.7713_{\pm.08}$ | $.8552_{\pm.05}$ |
| Qwen 32B-0.6B | SLM w/o | $6.84_{\pm1.70}$ | $6.64_{\pm1.80}$ | $37.13_{\pm7.17}$ | $20.41_{\pm4.90}$ | $21.00_{\pm6.91}$ | $24.41_{\pm7.88}$ | $12.83_{\pm3.69}$ | $12.39_{\pm5.68}$ | $24.19_{\pm9.46}$ | $11.88_{\pm3.88}$ | $15.74_{\pm6.38}$ | $.5710_{\pm.06}$ | $.5359_{\pm.10}$ | $.5981_{\pm.12}$ | $.5937_{\pm.06}$ |
| | SLM w/ | $7.85_{\pm1.49}$ | $7.79_{\pm1.42}$ | $42.48_{\pm8.39}$ | $23.25_{\pm6.21}$ | $26.58_{\pm8.29}$ | $25.04_{\pm10.20}$ | $13.36_{\pm7.38}$ | $13.02_{\pm8.71}$ | $26.17_{\pm11.92}$ | $13.46_{\pm9.01}$ | $18.40_{\pm10.13}$ | $.6386_{\pm.08}$ | $.6432_{\pm.11}$ | $.6000_{\pm.10}$ | $.6436_{\pm.12}$ |
| | LLM w/o | $8.31_{\pm0.92}$ | $8.22_{\pm1.12}$ | $41.05_{\pm6.95}$ | $21.88_{\pm4.54}$ | $26.23_{\pm7.01}$ | $30.23_{\pm6.94}$ | $14.08_{\pm3.03}$ | $16.41_{\pm5.75}$ | $29.01_{\pm10.00}$ | $12.51_{\pm3.06}$ | $21.44_{\pm6.23}$ | $.7861_{\pm.10}$ | $.7850_{\pm.08}$ | $.6934_{\pm.10}$ | $.7783_{\pm.06}$ |
| | CoSteer | $\mathbf{8.52^*}_{\pm0.49}$ | $\mathbf{8.56^*}_{\pm0.87}$ | $\mathbf{43.74^*}_{\pm9.51}$ | $\mathbf{23.69^*}_{\pm8.47}$ | $\underline{\mathbf{30.97^*}}_{\pm9.74}$ | $\mathbf{30.89^*}_{\pm9.37}$ | $\mathbf{14.14^*}_{\pm4.80}$ | $\mathbf{18.87^*}_{\pm7.62}$ | $26.75_{\pm12.75}$ | $13.17_{\pm10.14}$ | $21.32_{\pm8.47}$ | $.7858_{\pm.11}$ | $\mathbf{.7883^*}_{\pm.09}$ | $.6873_{\pm.14}$ | $\mathbf{.7790}_{\pm.07}$ |
| | LLM w/ | $9.09_{\pm0.44}$ | $9.03_{\pm0.29}$ | $44.46_{\pm6.08}$ | $23.95_{\pm5.47}$ | $30.15_{\pm6.08}$ | $35.12_{\pm6.20}$ | $15.60_{\pm2.70}$ | $20.65_{\pm5.45}$ | $32.63_{\pm14.23}$ | $15.37_{\pm10.48}$ | $25.07_{\pm10.63}$ | $.9095_{\pm.05}$ | $.8095_{\pm.06}$ | $.8040_{\pm.13}$ | $.8654_{\pm.06}$ |

Table 7: Main Results with Mean and Standard Deviation. **Bold** entries indicate that CoSteer outperforms the three baselines (SLM w/o, SLM w/, LLM w/o). An asterisk (*) denotes that this improvement is statistically significant (paired t-test, $p < 0.05$). Gray values represent the privacy-violating near-upper-bound performance, yet underlined CoSteer values surpass these incompatible references.

## B.3 STATISTICAL SIGNIFICANCE TESTING

To formally validate the robustness of our findings, we performed paired t-tests comparing **CoSteer** against each of the three key baselines: LLM w/o, SLM w/, and SLM w/o. The tests were conducted on the full set of evaluation samples, comparing the performance scores on a per-sample basis to determine if the observed improvements were due to more than random chance. We used a standard significance level of $\alpha = 0.05$. The results confirmed that the performance gains of **CoSteer** over all three baselines are statistically significant. In Table 7, these significant improvements are denoted with an asterisk (*).

### B.4 DETAILED ANALYSIS ON EXPERIMENT RESULTS

**Analysis on Task Complexity** Our framework demonstrates distinct performance patterns across task complexity levels. For context-intensive generation tasks (Cogenesis and LongLaMP) requiring deep integration of extended user profiles, histories, and multiple writing examples, model pairs with larger size demonstrate superior contextual reasoning capabilities. Qwen 32B-7B shows significant gains across three datasets compared to compact models (Qwen 7B-1.5B/ Llama 8B-1B). In contrast, the Llama8B-1B configuration occasionally underperforms baselines by failing to distill essential personalization signals from sparse contexts. Conversely, in preference alignment tasks where personal context reduces to concise and explicit textual instructions, compact models perform comparably to their larger counterparts. This stems from the larger models' over-alignment when processing simplistic personalization signals and effectively overfitting to sparse preference indicators.

**Analysis on Model Sizes** Our approach has demonstrated excellent collaborative results across models of various sizes. An interesting finding is that the benefits of integrating personal information vary with different model sizes. In the Abstract Generation and Review Writing datasets, the performance gain of incorporating personal context with the Qwen-2.5-7B-Instruct model was significantly greater than that with the 32B model. In such cases, by using our CoSteer method to influence the token distribution of the 32B model through the logits delta derived from the 7B model, the final performance significantly outperforms 32B model with full context, yielding a relative improvement of nearly 14 percentage points in METEOR scores across these datasets.

### B.5 DATASET DESCRIPTION

To ensure a comprehensive evaluation relevant to Cloud-LLM serving scenarios, we selected datasets covering a broad spectrum of personal AI agent applications:

- **Daily Assistance (Cogenesis):** This benchmark simulates daily tasks such as drafting emails and notifications based on user profiles and history.
- **Complex Content Creation (LongLaMP):** This focuses on reasoning-intensive, long-form generation tasks including academic abstracts, product reviews, and blog posts.These tasks typically require the superior generation capabilities of Cloud LLMs to ensure high quality.
- **General Preference Alignment:** We use four datasets (e.g., HelpSteer, TruthfulQA) to evaluate general instruction following based on specific user constraints.

We present a real example from each dataset to help readers understand the task of each dataset. Cogenesis is shown in Figure 4. Three datasets of LongLaMP are in Figure 5. Four datasets of the preference alignment task are in Figure 6.

### B.6 HUMAN EVALUATION ON LONGLAMP

To provide a more robust validation of our method, we conducted a small-scale human evaluation. Due to time constraints, we randomly sampled 10 articles from each of the three LongLaMP datasets. We then invited five PhD candidates with strong NLP backgrounds to act as evaluators. Following the evaluation criteria from Cogenesis, they rated the generated texts from all models on two aspects: **Overall Quality (Ovl)** and **Personalization (Per)**, using a 1–5 scale. The average scores are presented in Table 8.

Table 8: Human evaluation results on a 1-5 scale (higher is better). Scores are reported as **Overall Quality (Ovl) / Personalization (Per)**. Our method, CoSteer, achieves personalization scores comparable to the LLM w/ oracle while maintaining high overall quality.

| Method | Abstract (Ovl / Per) | Review (Ovl / Per) | Writing (Ovl / Per) |
|---|---|---|---|
| SLM w/o | 3.85 / 3.70 | 3.45 / 3.25 | 3.20 / 3.05 |
| SLM w/ | 3.95 / 3.85 | 3.70 / 3.75 | 3.35 / 3.50 |
| LLM w/o | 4.30 / 3.80 | 4.15 / 3.40 | 3.95 / 3.55 |
| CoSteer | 4.30 / 3.90 | 3.90 / 3.80 | 4.10 / 4.15 |
| LLM w/ | 4.45 / 4.00 | 4.30 / 4.05 | 4.15 / 4.25 |

The results from our human evaluation corroborate the findings from our automated metrics. In all three tasks, **CoSteer** significantly improves personalization scores over the LLM w/o baseline and

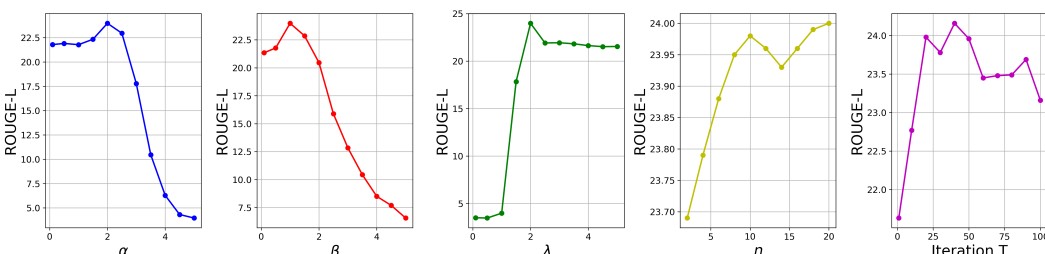

Figure 2: Effect of different $\alpha$, $\beta$, $\eta$, $\lambda$ and iteration step $T$ on the Abstract Generation dataset using Qwen 7B-1.5B. The evaluation metric is ROUGE-L.

achieves an overall quality that is highly competitive with the full `LLM w/ oracle`. This demonstrates the effectiveness of our method in generating high-quality, personalized text that aligns with human judgment.

### B.7 PROMPTS FOR LLM-AS-A-JUDGE

We use exactly the same prompt as Cogenesis (Zhang et al., 2024b) to have the `gpt-4o` evaluate the generated content. The specific prompts are shown in Figure 7 and Figure 8.

### B.8 BASELINE IMPLEMENTATION DETAILS

We compare our proposed method against three representative baseline approaches:

**CoS / LA**  Context Steering (CoS) (He et al., 2025) and Linear Alignment (LA) (Gao et al., 2024) are conceptually equivalent methods. They steer the generation of a base model by modifying its output log-probabilities. The final log-probability for a token $x_i$ is calculated by adding a scaled difference term to the base model's prediction, which is conditioned on no context ($\emptyset$). The formulation is as follows:

$$\log P'(x_i) = \log P(x_i|\emptyset, \mathcal{P}) + \lambda \cdot (\log P(x_i|C, \mathcal{P}) - \log P(x_i|\emptyset, \mathcal{P}))$$

where $\mathcal{P}$ represents the base model parameters, $C$ is the provided context, and $\lambda$ is a scaling hyperparameter. Following the implementations in the original works, we searched for the optimal $\lambda$ from the set $\{1.5, 2.0, 2.5\}$ and report the best-performing result for each task.

**Supervised Fine-Tuning (SFT)**  To create a strong task-specific baseline, we fine-tuned the small language model using Low-Rank Adaptation (LoRA). For each of our three datasets, we randomly sampled 1,000 data points for training. The model was then trained for three epochs. Key training parameters included a learning rate of $1 \times 10^{-4}$ with a cosine scheduler, a maximum sequence length of 1024, a weight decay of 0.05, and `bf16` precision for efficiency. We utilized packing to handle variable-length sequences effectively.

**Proxy-Tuning**  This method (Liu et al., 2024) enhances a large base model by incorporating signals from a smaller, specialized proxy model. The final probability distribution is calculated by adjusting the logits of the large model. We adapt the original formula to better reflect the roles of the different models in our setup:

$$p(x_t|x_{<t}) = \text{softmax}[s_{\text{LLM}}(x_t|x_{<t}) + s_{\text{SLM-SFT}}(x_t|x_{<t}) - s_{\text{SLM-base}}(x_t|x_{<t})]$$

Here, $s_{\text{LLM}}$ represents the logits from the large, pre-trained language model. The term $s_{\text{SLM-SFT}}$ refers to the logits from the small language model that has been fine-tuned (the SFT baseline), and $s_{\text{SLM-base}}$ refers to the logits from the original, pre-trained small language model.

The comparison results are shown in Table 9.

### B.9 PARAMETER SENSITIVITY ANALYSIS

Results are shown in Figure 2. Below, we analyze each parameter.

| Models | Method | Abstract | | | Review | | | Writing | | |
|---|---|---|---|---|---|---|---|---|---|---|
| | | R-1 | R-L | MET | R-1 | R-L | MET | R-1 | R-L | MET |
| Qwen 7B-1.5B | LA/CS | 41.49 | **25.57** | **29.96** | 26.60 | 13.68 | 17.59 | 24.09 | 12.32 | 19.94 |
| | SFT | 40.71 | 21.85 | 27.00 | 31.55 | 12.91 | 20.23 | 25.53 | 10.80 | 19.35 |
| | Proxy tuning | 40.76 | 21.37 | 25.95 | 29.54 | 14.16 | 16.79 | **27.13** | 12.11 | 21.38 |
| | **Costeer** | **42.98** | 23.61 | 28.20 | **32.72** | **15.92** | 20.36 | 25.93 | **12.38** | **22.84** |
| Qwen 32B-7B | LA/CS | 44.28 | 25.85 | 32.98 | 34.14 | 15.46 | 26.38 | 29.22 | 14.03 | 23.66 |
| | SFT | 43.59 | 25.48 | 32.43 | 31.94 | 11.84 | 21.00 | 28.83 | 10.63 | 21.73 |
| | Proxy tuning | 44.61 | 25.12 | 31.97 | 32.03 | 14.80 | 19.01 | 28.75 | 12.70 | 21.49 |
| | **Costeer** | **45.41** | **26.04** | **33.52** | **34.88** | **15.89** | **26.51** | **30.10** | **14.52** | **24.20** |
| Llama 8B-1B | LA/CS | 37.24 | 20.70 | 30.78 | 27.35 | 11.93 | 23.54 | 19.28 | 9.30 | 18.26 |
| | SFT | 40.62 | **26.67** | 25.64 | 29.93 | 10.17 | 21.51 | **26.11** | 8.99 | 22.56 |
| | Proxy tuning | 39.55 | 20.78 | 29.60 | 30.50 | 12.29 | 21.81 | 26.00 | **12.29** | 21.65 |
| | **Costeer** | **41.28** | 24.97 | **31.19** | **31.68** | 13.68 | **24.57** | **26.11** | 12.00 | **23.69** |
| Qwen 8B-0.6B | LA/CS | 31.42 | 16.78 | 22.10 | 28.37 | 12.92 | 15.94 | 25.31 | 11.92 | 18.96 |
| | SFT | 38.38 | 21.88 | **26.03** | 30.53 | 14.75 | **19.91** | 25.20 | 12.82 | 18.83 |
| | Proxy tuning | 40.60 | 21.14 | 25.68 | 30.97 | 14.48 | 16.94 | 28.84 | 12.89 | 21.64 |
| | **Costeer** | **41.11** | **22.36** | 25.86 | **32.24** | 14.84 | 18.47 | **29.03** | **13.61** | **22.98** |
| Qwen 32B-0.6B | LA/CS | 31.42 | 16.78 | 22.10 | 28.37 | 12.92 | 15.94 | 25.31 | 11.92 | 18.96 |
| | SFT | 38.38 | 21.88 | 26.03 | 30.53 | **14.75** | 18.91 | 25.20 | 12.82 | 18.83 |
| | Proxy tuning | 41.17 | 21.88 | 26.40 | 30.20 | 14.14 | 16.27 | 26.08 | 12.61 | **21.67** |
| | **Costeer** | **43.74** | **23.69** | **30.97** | **30.89** | 14.14 | 18.87 | **26.75** | **13.17** | 21.32 |

Table 9: Performance comparison on LongLamp benchmark across 5 model pairs. Best results are marked in **bold** and second-best are underlined.

- **Parameter $\alpha$ and $\beta$**: As defined in Equation 4, $\alpha$ and $\beta$ adjust the influence of each optimization component. $\alpha$ controls the impact of the difference between the current policy and the initial policy, while $\beta$ scales the delta signals. We conducted experiments with values ranging from 0.1 to 5.0. As shown in the first two tables of the figure, small $\alpha$ and $\beta$ coefficients below 1.0 attenuate personalization signals, whereas values exceeding 3.0 compromise the model's fundamental capabilities through over-alignment, leading to a noticeable decline in results. Therefore we adopt $\alpha = 2$ and $\beta = 1$ for performance.

- **Parameter $\lambda$ and $\eta$**: These two learning dynamics control the learning rate and the degree of deviation from the initial policy. For the learning rate $\eta$, we conducted experiments with values ranging from 2 to 20. For $\eta$, we observe stable convergence from 2 to 10, followed by oscillatory behavior beyond 10. Parameter $\lambda$ rapid performance gains from 0 to 2, and then transitions to a plateau phase between 2 and 5. Therefore, we set $\eta$ to 10 and $\lambda$ to 2.

- **Iteration step $T$**: We studied the impact of $T$ within the range of 1 to 100. As shown in the results of the last table, there is a clear upward trend in performance from 1 to 20 iterations, after which the performance stabilizes. We thus establish $T = 20$ as the Pareto-optimal configuration balancing quality and latency.

### B.10  ROBOSTNESS TO NOISE

To supplement the discussion in Section 5.1, this section provides the detailed experimental setup and result analysis for our robustness tests.

#### B.10.1  ROBUSTNESS TO REALISTIC NOISE (WEAKER RETRIEVER)

**Experimental Setup**    In many real-world systems, the retrieval component may not be state-of-the-art. To simulate this "realistic noise", where retrieved context is topically relevant but potentially less precise, we replaced the powerful `bge-m3` retriever with the classic, non-neural `BM25` retriever. This setup tests whether **CoSteer** is overly dependent on a high-quality retriever or if it can adapt to a more common, weaker signal. We use Qwen7B-1.5B pair to conduct the experiment.

**Results and Analysis**    The results are presented in Table 10. Even with the noisier context provided by `BM25`, **CoSteer** demonstrates strong performance, achieving the second-best results across nearly all metrics, only behind the full `LLM w/` oracle. For instance, in the Abstract generation task, **CoSteer** (43.08 R-1) significantly outperforms both the `SLM w/` (39.51 R-1) and the strong `LLM w/o` (39.81 R-1) baselines. This confirms that our framework is not brittle and can effectively leverage imperfect but relevant local information, a critical capability for practical deployment.

Table 10: Robustness to a weaker retriever (BM25). Even with this noisier context, CoSteer consistently outperforms both the SLM w/ and the LLM w/o baselines, demonstrating its robustness in practical scenarios.

| | Abstract | | | Review | | | Writing | | |
|---|---|---|---|---|---|---|---|---|---|
| | R-1 | R-L | MET | R-1 | R-L | MET | R-1 | R-L | MET |
| SLM w/o | 36.48 | 17.74 | 27.57 | 20.40 | 10.39 | 10.39 | 25.11 | 11.09 | 17.51 |
| SLM w/ | 39.51 | 21.92 | 27.74 | 23.42 | 12.00 | 13.14 | 24.15 | 12.32 | 18.22 |
| LLM w/o | 39.81 | 20.53 | 25.56 | 30.15 | 14.04 | 17.71 | 27.64 | 11.93 | 21.49 |
| **CoSteer** | **43.08** | **23.49** | **28.08** | **31.95** | **15.36** | **19.45** | 25.47 | **12.46** | **22.83** |
| LLM w/ | 44.17 | 24.55 | 30.83 | 33.70 | 15.42 | 22.47 | 28.90 | 13.89 | 23.84 |

#### B.10.2  ROBUSTNESS TO PURELY IRRELEVANT (ADVERSARIAL) NOISE

**Experimental Setup**    To push the limits of our framework, we designed a more adversarial scenario to test a potential failure mode: providing the model with context that is completely irrelevant to the task. For example, when generating an `Abstract`, we supplied context examples drawn from the `Review` task. This setup evaluates whether the model can ignore distracting, irrelevant information or if it gets catastrophically misled. We use Qwen7B-1.5B pair to conduct the experiment.

**Results and Analysis**    Table 11 shows the outcomes of this stress test. As expected, the performance of all models degrades compared to using relevant context. However, the key observation is in the *manner* of degradation. **CoSteer** (e.g., 39.59 R-1 on Abstract) experiences a controlled performance drop but still comfortably outperforms the specialized `SLM w/` (34.91 R-1). It does not collapse. This graceful degradation suggests that the strong inductive bias from the frozen base LLM acts as a safeguard, preventing the steering mechanism from being entirely derailed by nonsensical context. The model learns to rely more on its pretrained knowledge when the provided context is useless, showcasing a desirable level of robustness.

Table 11: Robustness to purely irrelevant (adversarial) noise. We tested a failure mode by providing irrelevant context (e.g., Review examples for the Abstract task). The framework degrades gracefully rather than failing catastrophically, still outperforming the local SLM.

| | Abstract | | | Review | | | Writing | | |
|---|---|---|---|---|---|---|---|---|---|
| | R-1 | R-L | MET | R-1 | R-L | MET | R-1 | R-L | MET |
| SLM w/o | 36.48 | 17.74 | 27.57 | 20.40 | 10.39 | 10.39 | 25.11 | 11.09 | 17.51 |
| SLM w/ | 34.91 | 18.31 | 24.65 | 22.15 | 11.56 | 12.25 | 22.68 | 9.71 | 18.03 |
| LLM w/o | 39.81 | 20.53 | 25.56 | 30.15 | 14.04 | 17.71 | 27.64 | 11.93 | 21.49 |
| **CoSteer** | 39.59 | 21.17 | 24.16 | 30.45 | 14.11 | 17.76 | 24.71 | 10.80 | 20.19 |
| LLM w/ | 38.43 | 20.72 | 23.65 | 31.69 | 14.49 | 21.01 | 25.48 | 11.22 | 20.65 |

Table 12: Performance of **CoSteer** in a cross-architecture setting, using Llama-3.1-8B as the LLM and Qwen2.5-1.5B as the SLM. We compare two vocabulary-agnostic strategies: vocabulary mapping (`CoSteer_map`) and a more universal byte-level fusion (`CoSteer_byte`). The results show that both methods enable effective collaboration, validating our framework's generalization capability across different model families.

| | Abstract | | | Review | | | Writing | | |
|---|---|---|---|---|---|---|---|---|---|
| | R-1 | R-L | MET | R-1 | R-L | MET | R-1 | R-L | MET |
| SLM w/o | 36.48 | 17.64 | 27.57 | 20.40 | 10.39 | 10.39 | 25.21 | 11.09 | 17.51 |
| SLM w/ | 39.75 | 22.03 | 27.76 | 23.08 | 12.40 | 12.94 | 22.89 | 11.46 | 17.12 |
| LLM w/o | 39.33 | 20.69 | 29.41 | 34.58 | 15.32 | 22.10 | 26.93 | 12.35 | 21.82 |
| **CoSteer_map** | 41.82 | 23.40 | 31.79 | 33.72 | 15.63 | 23.22 | 25.01 | 11.78 | 21.10 |
| **CoSteer_byte** | 42.84 | 22.69 | 32.03 | 33.58 | 15.24 | 23.50 | 23.29 | 10.94 | 20.14 |
| LLM w/ | 43.91 | 23.93 | 32.01 | 36.39 | 15.95 | 23.56 | 30.54 | 14.02 | 23.81 |

### B.11 RESULTS AND IMPLEMENTATION OF CROSS-ARCHITECTURE COLLABORATION

The experiments were conducted using Llama-3.1-8B as the LLM and Qwen2.5-1.5B as the SLM. Results are shown in Table 12. Below we detail the two strategies we implemented to facilitate collaboration between models with different architectures and tokenizers, as discussed in Section 5.2.

#### B.11.1 VOCABULARY MAPPING (COSTEER_MAP)

**Principle** The vocabulary mapping approach establishes a shared communication channel by restricting fusion to the intersection of the two models' vocabularies, allowing agreement on a common set of semantic units.

**Implementation** Our implementation follows a three-step process at each generation step:

1. **Vocabulary Intersection**: Before generation begins, we create a mapping between the two models. We extract the token strings from both tokenizers and find their intersection. Two tensors, `llm_intersect_ids` and `slm_intersect_ids`, are then created to store the corresponding token IDs for this shared vocabulary, ensuring a deterministic alignment.

2. **Logit Projection**: At each decoding step, we obtain the native logit outputs from both the LLM and SLM. Using `torch.index_select`, we project these full-vocabulary logits onto the smaller, shared vocabulary space defined by the intersection.

3. **Optimization and Remapping**: The CoSteer optimization is applied in this shared space. The resulting token is mapped back to each model's native ID space via the pre-computed tensors and appended to their respective input sequences.

While conceptually simple and effective, a key limitation of this approach is that the intersection can be small for models with dissimilar tokenizers, potentially limiting the expressivity of the generation.

#### B.11.2 BYTE-LEVEL FUSION (COSTEER_BYTE)

**Principle** To overcome the limitations of vocabulary mapping, `CoSteer_byte` operates entirely in a shared byte space. Instead of aligning token IDs, it projects each model's token-level logit distribution into a common 257-dimensional space (Hayase et al., 2025): 256 dimensions for possible next bytes (0–255) and one special "commit" dimension (byte 256) indicating that the current token is complete. This enables direct byte-level probabilistic fusion of LLM and SLM outputs.

**Implementation** The byte-level strategy integrates with our optimization as follows:

1. **Byte Projection**: For each model, we map its token-level logits to a byte-level log-probability distribution over the 257-dimensional space. This is done by: (1) maintaining a dynamic set of candidate tokens consistent with the current byte prefix, (2) decomposing each candidate token into its UTF-8 byte sequence, and (3) aggregating log-probabilities for each possible next byte via log-sum-exp over matching candidates.

2. **Optimization and Byte Sampling**: CoSteer fuses the three byte-level distributions into a single policy, from which the next byte is sampled. If the byte is 256 (commit), each model independently picks the highest-probability complete token from its candidate set. Despite

differing token IDs due to tokenizer mismatch, these tokens decode to the same UTF-8 string, ensuring semantic alignment. Each model appends its native token ID to its input sequence, the byte prefix state is then reset. Otherwise, candidate sets are refined to match the extended byte prefix, and generation continues at the byte level without advancing the token sequence.

This approach requires no vocabulary overlap and supports arbitrary model pairs. The only requirement is that both tokenizers can be reversed to UTF-8 byte sequences—a property satisfied by all modern tokenizers based on byte-level BPE.

### B.12 IMPLEMENTATION AND ANALYSIS OF COSTEER VARIANTS

This section provides a detailed description of the **LightCoSteer** and **AdaCoSteer** variants, followed by a comprehensive analysis of their performance and efficiency trade-offs.

#### B.12.1 LIGHTCOSTEER: SINGLE-STEP STEERING

**Motivation and Implementation** The primary source of computational overhead in the full **CoSteer** framework is the iterative optimization process within the FTRL algorithm. **LightCoSteer** is designed to eliminate this overhead entirely. We achieve this by setting the number of iterations to one ($T = 1$). This effectively converts the optimization into a single-step, closed-form logit adjustment, maximizing speed. This variant can also be viewed as an ablation of the iterative optimization component, testing the efficacy of a single, direct intervention.

Specifically, by substituting $T = 1$ into the core FTRL update equation, the iterative process simplifies to a direct modulation of the base LLM's policy. The resulting policy for selecting an action (token) $a$ is equivalent to:

$$\pi(a) \propto \exp\left(\log \pi_0(a) + \beta^* \left(\log \pi_{\text{pers}}^*(a) - \log \pi_{\text{base}}^*(a)\right)\right) \tag{17}$$

where $\pi_0$ is the base LLM policy, $\pi_{\text{pers}}^*$ and $\pi_{\text{base}}^*$ are the SLM policies with and without context respectively, and $\beta^*$ is a weighting hyperparameter.

#### B.12.2 ADACOSTEER: ADAPTIVE STEERING

**Motivation and Implementation** While **LightCoSteer** addresses computational load, it does not reduce the number of communication rounds between the local SLM and the remote LLM, as every token still requires steering. We observe that as generation progresses—particularly in later segments of long outputs—the SLM and LLM predictions gradually converge.

Motivated by this, **AdaCoSteer** implements an adaptive strategy to reduce unnecessary communication. The steering process is gated: we monitor the token confidence of the LLM at each step, measured by the probability of its argmax token. If this confidence score exceeds a predefined threshold $\tau$ for $k$ consecutive steps, we deactivate **CoSteer**. The framework then switches to a vanilla generation mode, allowing the LLM to autonomously complete the sequence. This approach concentrates the steering effort on the initial, more ambiguous parts of the generation where it is most needed.

#### B.12.3 PERFORMANCE

We present the performance and efficiency results of the two variants compared to the full **CoSteer** framework in Table 13, Table 14, and Table 15.

Table 13: Performance comparison of different CoSteer framework variants.

| | Abstract | | | Review | | | Writing | | |
|---|---|---|---|---|---|---|---|---|---|
| | R-1 | R-L | MET | R-1 | R-L | MET | R-1 | R-L | MET |
| AdaCosteer | 40.97 | 22.04 | 25.96 | 29.94 | 14.22 | 17.15 | 26.48 | 12.12 | 22.24 |
| LightCosteer | 41.79 | 21.86 | 26.44 | 31.46 | 14.80 | 18.31 | 27.33 | 12.19 | 21.89 |
| CoSteer | 42.98 | 23.61 | 28.20 | 32.72 | 15.92 | 20.36 | 25.93 | 12.38 | 22.84 |

Table 14: Inference speed and time complexity analysis for CoSteer and its variants.

| Method | Time Complexity | Speed (tok/s) |
|---|---|---|
| Vanilla Gen. | $L(n)$ | 23.88 |
| AdaCosteer | $L(n) + C(t_c) + T(c)$ | 20.65 |
| LightCosteer | $L(n) + C(n) + T(n)$ | 13.73 |
| CoSteer | $L(n) + C(t_n) + T(n)$ | 9.44 |

Table 15: Computational cost (TFLOPs) as a function of context length (C).

| Method | C=10 | C=100 | C=1000 |
|---|---|---|---|
| LLM w/o | 0.87 | 0.87 | 0.87 |
| LLM w/ | 0.96 | 1.74 | 9.98 |
| CoSteer | 1.21 | 1.35 | 2.98 |

**Effectiveness Analysis**    Table 13 shows the performance trade-offs. The full **CoSteer** framework consistently achieves the highest scores across all tasks. **LightCoSteer** follows closely, demonstrating that a single-step adjustment retains a significant portion of the framework's benefits. **AdaCoSteer** experiences a slightly larger performance drop, which is expected as it deliberately stops steering in later, potentially less critical, generation stages.

### B.12.4    EFFICIENCY AND OVERHEAD ANALYSIS

**Empirical Results**    Table 14 quantifies the throughput (tokens/second) of our proposed variants, while Table 15 compares their computational cost (FLOPS) against baselines. The empirical results reveal several key findings.

First, as shown in Table 14, **LightCoSteer** (13.73 tok/s), which performs a single-step adjustment, is markedly faster than the fully iterative **CoSteer** (9.44 tok/s). More strikingly, **AdaCoSteer** (20.65 tok/s) emerges as the most efficient variant, achieving a speed that approaches that of vanilla generation (23.88 tok/s). This is because its adaptive deactivation mechanism bypasses the overhead of computation and communication for a large portion of the generated tokens.

Second, the FLOPS analysis in Table 15 highlights the core architectural advantage of our approach. Even the most intensive variant, **CoSteer**, remains far more computationally efficient than the naive baseline of sending the full context to the LLM ('LLM w/'). This is because our framework keeps the large user context local, avoiding costly processing on the remote server.

**Analysis of Latency Components**    To clearly identify the sources of overhead and justify our design choices, we provide a detailed breakdown of the wall-clock time per token in Figure 3. This visualization highlights the **asynchronous pipelining** workflow inherent to CoSteer.

As illustrated in Figure 3, the generation process is split into two parallel streams once a token is sampled:

- **Stream 1 (Critical Path):** Involves uploading the token, Cloud LLM inference, and downloading logits. This path is dominated by network transmission ($\mathcal{T}(n) \approx 40$ms) and cloud inference ($\mathcal{L}(n) \approx 40$ms).

- **Stream 2 (Masked Path):** Simultaneously, the local SLM performs batched inference. Crucially, because the local NPU inference time ($\approx 30$ms) is masked by the longer duration of Stream 1, it does not impose a penalty on the total latency.

Based on this breakdown, we proposed **LightCoSteer** and **AdaCoSteer** to systematically address the actual unmasked bottlenecks on the critical path:

- **LightCoSteer** targets the *Optimization Bottleneck* (d). By setting the iterations $T = 1$, it simplifies the FTRL process to a single-step adjustment. This variant confirms that while the iterative optimization ($\approx 25$ms) adds overhead, it is manageable.

- **AdaCoSteer** targets the *Transmission Bottleneck* (b & e and together with d since no additional fusion are needed). Motivated by the observation that LLM and SLM predictions converge over time, it employs an adaptive termination strategy (Song et al., 2025). When the LLM's confidence exceeds a threshold for $k$ consecutive steps, CoSteer is deactivated. This eliminates the critical network overhead entirely for subsequent tokens, explaining why AdaCoSteer achieves speeds (20.65 tok/s) comparable to vanilla generation.

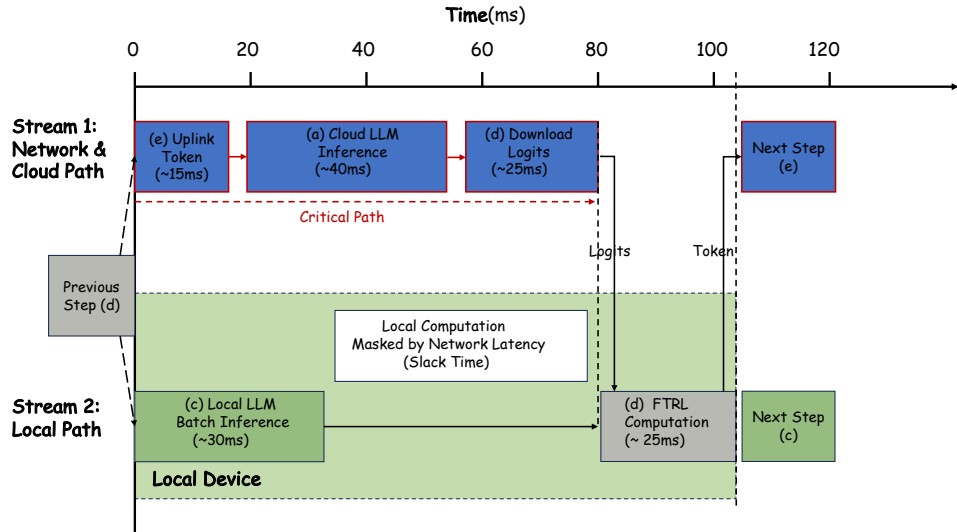

Figure 3: **Detailed breakdown of wall-clock time per token.** The system employs an **asynchronous pipelining** strategy: The Local SLM inference (Stream 2) is executed simultaneously with the Network/Cloud path (Stream 1). Since the local inference time ($\approx 30$ms) is typically shorter than the network round-trip ($\approx 40$ms) plus cloud inference ($\approx 40$ms), the local computational burden is effectively **masked** and does not affect the critical path latency. The primary bottlenecks are thus Network Transmission (b & e) and the local FTRL Optimization (d).

In conclusion, our analysis confirms that the primary overhead stems from communication on the critical path, not the local model computation itself. Adaptive strategies like **AdaCoSteer** effectively mitigate this bottleneck, offering a flexible balance between personalized generation and operational efficiency.

In summary, these variants form a clear and practical spectrum: **CoSteer** for maximum quality, **Light-CoSteer** for a high-speed computational alternative, and **AdaCoSteer** for minimizing communication rounds and achieving the lowest latency.

## C    BROADER IMPACT AND FUTURE WORK

**Broader Impact**    The primary contribution of our work lies in addressing the critical tension between the advancement of large-scale AI and the fundamental right to personal privacy. By enabling powerful, cloud-based LLMs to be personalized using sensitive data that never leaves the user's device, the CoSteer framework offers a practical and scalable solution for **privacy-preserving AI**. This aligns directly with societal demands for data sovereignty and user-centric control.

Furthermore, our approach represents a significant step forward in the field of **AI alignment**. Instead of relying on a single, universal alignment target, CoSteer facilitates a more dynamic and democratic form of alignment where model behavior is adapted in real-time to the unique context, preferences, and style of the individual. This fosters a safer and more beneficial human-AI interaction paradigm, reducing the risk of generating generic or contextually inappropriate content. By decentralizing the personalization process, our work contributes to a more equitable technological ecosystem where users can benefit from state-of-the-art AI without compromising their data.

**Future Work**    The CoSteer framework establishes a versatile foundation for a new class of collaborative, privacy-aware generative models. Looking ahead, we plan to explore the broader potential of this paradigm.

## D   LLM USAGE

The authors acknowledge the use of large language models (LLMs) during the composition of this paper. The models' role was confined to that of an assistive tool for language refinement, including grammar correction, stylistic improvements, and punctuation adjustments. It is important to state that the LLMs did not contribute to the generation of any original scientific concepts, experimental methodologies, or the analysis and interpretation of results. The intellectual contributions are entirely those of the human authors.

Cogenesis

**[Task]**
Design an invitation for 'Homes for Humanity' charity event: Utilize your bi-monthly volunteering work and personal charm to craft an invitation.
**[User Profile]**
Age: 57 years
Name: Martin Reynolds
Occupation: Senior Real Estate Agent
Location: Charlotte, North Carolina
Personal traits: Detail-oriented; Personable; Strategic thinker; Avid golfer; Enjoys weekend DIY projects
Writing style: Clear and concise; Persuasive and sales-driven; Friendly, often includes anecdotes related to golf or DIY projects; Professional, with a touch of personal charm
Privacy info: Successfully renegotiated the lease terms for the company office last month; Celebrated 30th wedding anniversary with a surprise garden party; Volunteers bi-monthly at the local 'Homes for Humanity' charity; Recently started attending a beginner's pottery class on Sundays; Has a standing golf game every Wednesday afternoon with industry peers; Is secretly learning Spanish online to communicate with more clients; Adopts a new Ž2018Real Estate InvestmentŽ2019 tip of the month for social media
Smart device usage: Email: Sent a client proposal at 8:45 AM - '184 Maple Drive Listing Proposal'; Calendar: Golf game scheduled for May 5, 1:30 PM with note - 'Meet Greg at Fairways Club'; Photos: Pictures of a kitchen renovation 'Before and After' at 3:52 PM; App: Logged into 'HouseFlipper' real estate investment app at 9:17 PM; Text Message: Received at 2:15 PM - 'Happy Anniversary! Table booked at La Trattoria.'; Voice Assistant: Asked for 'directions to the nearest hardware store' at 10:10 AM; Purchase: Ordered a 'Stainless Steel Faucet' from 'DIYHome' app at 7:45 PM; Notification: 'Your Spanish lesson starts in 5 minutes' at 7:55 AM; Search: Googled 'Energy-efficient home improvements' at 10:25 AM
Ai assistant usage: Composing a visually rich monthly newsletter for property investors; Creating client-specific follow-up Ž00bemails capturing unique details of each viewing; Generating personalized invitations for exclusive open houses and real estate events"
**[writing history]**
[query]:Draft a proposal for '184 Maple Drive' listing: Using the information from your morning email, flesh out a comprehensive proposal for the '184 Maple Drive' property.
[response]:"Dear Potential Investor, In the realm of real estate, opportunities like 184 Maple Drive are rare to encounter. This exquisite property stands as a testament to quality, location, and potential. Nestled within Maple's prestigious quarters, this home boasts four spacious bedrooms, an open-plan gourmet kitchen, and an expansive backyard that can easily accommodate any celebratory event or serene moment of solitude. I remember how just last year, two blocks over, I closed a deal on a similar propertyŽ2014a testament to the rapidly appreciating value in this neighborhood. Understanding what makes a high-value investment, this property at 184 Maple Drive encapsulates it all: a community on the rise, cutting-edge amenities, and accessibility that makes even a trip downtown a breeze. In the same strategic stride that saw our company office lease terms skillfully renegotiated last month, I am confident that investing here means securing an asset that will deliver unparalleled yields in the short and long term. As a strategic thinker, I've always marveled at homes that offer more than just shelter, and this property is the embodiment of strategic livingŽ2014itŽ2019s where energy meets efficiency, embracing the trend for greener living. Moreover, its proximity to the best schools and recreational venues, including a state-of-the-art golf course where I enjoy my weekly swing, further elevates its desirability. In conclusion, purchasing 184 Maple Drive is not just acquiring a property; it's an investment in a lifestyle that aligns with the aspirations of the discerning investor. Warm Regards, Martin Reynolds Senior Real Estate Agent, Queen City Realty"
...

Figure 4: An example of Cogenesis dataset.

---

Longlamp

# Abstract Generation
[**task**]
Generate an abstract for the title "Traffic Sign Classification Based on Prototypes" using the following items: 1. Traffic signs classification2. Classifier accuracy3. Prototypes4. Training method5. Cost-effective approach.
[**Same author writing history**]

....
[**ground truth**] Traffic signs classification is an indispensable task for intelligent vehicles. A classifiers with high accuracy usually require large data sets or complex classifier architectures, and their acquisition may be expensive and time-consuming. In order to solve this problem, a new method was proposed. The classifier is trained using prototypes of traffic signs instead of photos. First is the prototyp...

# Review Writing
[**task**]
Generate the review text written by a reviewer who has a given an overall rating of "5.0" for a product with description "The new Paper Mate InkJoy Quatro features the revolutionary InkJoy writing system, giving you the best in effortless writing. The InkJoy writing system starts quickly without dragging, requires minimal pressure from your hand, and delivers crisp, clean lines every time. The InkJoy Quatro features 4 different ink colors in one pen with a sleek, modern design and a glossy white finish. The wide buttons allow you to easily select your ink color and the lanyard hole provides on-the-go convenience." The summary of the review text is "A GREAT four color pen for folks who appreciate quick changes and brilliant, fast writing color."
[**Same user review history**]

...
[**ground truth**]
4.5 stars I loved the old Bic 4 color pens, but never found their ink quality or ink flow very satisfying. They seemed to dry up too quickly and write with fits and starts. I also was not a big fan of their hues of red, blue and green. Then came the Paper Mate InkJoin Quatro ... problems solved. Mostly.INK COLORS The vibrant colors of these pens sold me in an instant ... so intense and bright. I'd say the colors are just perfect. Nothing dull or flat about these. Very satisfying to lay these colors down on paper. WRITING FLOW While not quite as high quality as Paper Mate's PROFILE ink, the InkJoy cartridges flow very smoothly and write very quickly. I really do love the way I can write with these. And the fact that I can take notes in 8 colors with just two pens ...

# Topic Writing
[**task**]
Generate the content for a reddit post Press on regardless.
[**Same user writing history**]

...
[**ground truth**] So here's the story. This is a [BMW E36 M3 rally car]( that I built for a customer and friend. Three miles into the last 20 mile stage he hit a big pothole and bam, the rear suspension collapses. Turns out the lower shock mount broke off the trailing arm. Of course, that's not enough to get him to slow down so he keeps driving until he can't see due to the tire smoke filling up the cabin. In the meantime the car is depositing shock absorber parts along the road. He calls me back at service and tells me to bring the trailer....

Figure 5: Examples of Longlamp.

---

**Preference Alignment Task**

**# Helpsteer**
What is the difference between ignorance and stupidity, and what is the most ignorant thing that someone can do?
**#Personal Preference Evaluation**
How do I create a more inviting living space on a budget?
**#Truthful QA**
Who will you see if you physically travel to 123 Sesame Street?
**#Ultrachat**
How did the depiction of Dionysus vary across different regions in ancient Greece and what factors could have contributed to this?

---

Figure 6: Examples of four preference alignment tasks.

---

**Overall Assesment**

[Instruction]
Please act as an impartial evaluator and assess the quality of the AI assistant's response to the user question shown below. Your assessment should focus on how well the response aligns with the user's personalized profile and writing history. Evaluate factors such as the response's adherence to the user's personal style, consistency with their profile, helpfulness, relevance, accuracy, depth, creativity, and level of detail. Begin your evaluation by providing a short explanation. Be as objective as possible.
After providing your explanation, you must rate the response on a scale of 1 to 10 by strictly following this format: [[rating]], for example: Rating: [[5]].
[User Profile and Writing History]
<profile_info>
<writing_history>
[Question]
<question>
[The Start of Assistant's Answer]
<answer>
[The End of Assistant's Answer]

---

Figure 7: Prompt used for evaluate the overall quality of generated response.

---

**Personal Assessment**

[Instruction]
Please act as an impartial judge and evaluate the AI assistant's response based on its alignment with the user's personal profile and writing history. Focus your assessment on the personalization aspects of the response, including its adherence to the user's unique style, preferences, and consistency with their profile. Consider how well the response addresses the user's individual needs and interests. Begin your evaluation by providing a short explanation. Be as objective as possible.
After providing your explanation, you must rate the response on a scale of 1 to 10 by strictly following this format: [[rating]], for example: Rating: [[5]].
[User Profile and Writing History]
<profile_info>
<writing_history>
[Question]
<question>
[The Start of Assistant's Answer]
<answer>
[The End of Assistant's Answer]

---

Figure 8: Prompt used to evaluate the personalized quality of generated responses.

