# OpenReview forum: "CoSteer: Collaborative Decoding-Time Personalization via Local Delta Steering"
_ICLR.cc/2026/Conference — Submitted to ICLR 2026_

### Official Review · Reviewer_QsZr · 2025-10-25

**Soundness:** 3
**Presentation:** 3
**Contribution:** 3
**Rating:** 4
**Confidence:** 3

**Summary:**

This paper introduces CoSteer, a collaborative framework for personalized text generation, operating within a cloud-edge paradigm. It aims to leverage the power of large language models (LLMs) in the cloud while preserving user privacy by keeping sensitive context (profiles, preferences, history) strictly on the local device. The core challenge is enabling personalization without transmitting private data to the cloud, while avoiding the quality limitations of using only a small local model (SLM).

CoSteer proposes a decoding-time steering mechanism that is tuning-free. A local SLM computes the logit difference (delta) between its outputs generated with and without access to the private user context. This delta vector, representing the personalization direction, is calculated and applied locally to steer the logits received from the cloud LLM at each decoding step. The steering process uses an online learning formulation (FTRL) with an efficient closed-form update, ensuring privacy as only the final steered token is returned to the cloud.

**Strengths:**

Tackles the critical challenge of balancing LLM capabilities, user privacy, and personalization on resource-constrained devices, which is crucial for real-world applications like personal assistants. While cloud-edge collaboration for personalization isn't entirely new, CoSteer introduces a specific, elegant mechanism using the SLM's logits delta combined with an online learning (FTRL) formulation for steering. This particular approach to extracting and applying the personalization signal locally is a key technical contribution.

**Weaknesses:**

The core concept of cloud-edge collaboration or "semi on-device" processing to balance privacy, personalization, and compute power, is an active area of research. While the specific delta steering mechanism is novel, the overall collaborative architecture might be seen as an instantiation within a known paradigm rather than a completely groundbreaking framework. The per-token communication round trip (cloud-to-device for logits, device-to-cloud for token) remains a significant practical bottleneck, especially under high network latency. The paper acknowledges this and proposes AdaCoSteer, but a direct latency comparison with vanilla cloud LLM generation is missing. Requiring the local SLM to run twice per token (with/without context) plus the FTRL optimization step could impose a non-trivial computational and energy burden on the local device, potentially exceeding the cost of running the SLM just once.

**Questions:**

Can you provide a more detailed breakdown of the wall-clock time per token? Specifically, measurements for: (a) Cloud LLM inference, (b) Cloud-to-local logits transmission, (c) Local SLM inference (x2), (d) Local FTRL optimization (using Eq 7), (e) Local-to-cloud token transmission? How does the total per-token latency compare quantitatively to vanilla LLM cloud inference under typical network conditions?

Could you elaborate on the intuition behind why the iterative FTRL optimization (T=20) provides better results than the single-step LightCoSteer variant (T=1)? Does iteratively refining the policy within the generation of a single token allow for better integration of the SLM delta signal?

---

> ### Author Response · Authors · 2025-11-20
>
> Dear Reviewer QsZr,
>
> Thanks for your constructive review and for recognizing our CoSteer framework as a "**specific, elegant mechanism**" that tackles the "**critical challenge**" of balancing capability and privacy. Below, we provide point-by-point responses to address your concerns.
>
> ---
>
> **Regarding the concern about technical contribution:**
>
> We respectfully suggest that our contribution extends beyond a simple instantiation of a collaborative paradigm. While the concept of cloud-edge collaboration exists, CoSteer introduces specific system-level and algorithmic innovations to make *decoding-time steering* viable in this setting:
>
> * **Paradigm Shift:** We verify a novel "weak-to-strong" capability where a tiny SLM effectively steers a massive LLM.
> * **Novel Formulation:** We novelly define the utility function based on local logit deltas, repurposing online learning for distributed, privacy-preserving coordination.
> * **Scenario-Specific Designs:** We introduce **AdaCoSteer** to minimize latency and **Byte-Level Fusion** to resolve tokenizer mismatches.
>
> We do not seek to claim that CoSteer is "completely groundbreaking" in every aspect; rather, we take this opportunity to clearly delineate our specific technical contributions to make decoding-time steering viable in this setting.
>
> ---
>
> **Regarding the detailed latency breakdown and computational burden:**
>
> We sincerely thank you for requesting this breakdown. This gives us the opportunity to clarify our system's parallelized workflow and explain why the local computation is not a bottleneck.
>
> **1. Clarification on Computational Burden (Batching & Pipelining):**
>
> We respectfully clarify that the local SLM inference does **not** impose a latency penalty due to our parallelization strategies.
>
> * **Batch Inference:** The two SLM forward passes—one with and one without context—are executed in a single batch ($N=2$). On modern edge devices, the latency of processing a batch of two is only marginally higher than a batch of one, and significantly less than running them sequentially.
>
> * **Asynchronous Pipelining:** Furthermore, we employ asynchronous pipelining to effectively mask this local process behind the network latency. Specifically, once a token is sampled, the system operates in two **simultaneous** streams:
>
>     * *Stream 1 (Network Path):* The token is uploaded (e), processed by Cloud LLM (a), and logits are downloaded (b).
>
>     * *Stream 2 (Local Path):* Simultaneously, the local SLM performs batched inference (c).
>
> Upon the completion of both paths, these streams converge locally to execute the FTRL optimization (d) for final token sampling. Since the inference of the tiny local model is typically faster than the combined duration of the cloud round-trip and processing, it does not affect the critical path latency.
>
> **2. Estimated Wall-Clock Time per Token:**
>
> Based on our system profiling, the wall-clock time per token breaks down as follows:
>
> * **(a) Cloud Inference:** $\approx$ **40ms** (Fixed cost).
> * **(b) & (e) Network Transmission:** $\approx$ **40ms** (Primary bottleneck). Downlink logits **(b)** dominate ($\approx$ **25ms**); uplink token **(e)** is minimal ($\approx$ **15ms**).
> * **(c) Local SLM Inference:** $\approx$ **30ms** (**Masked** by cloud path).
> * **(d) Local FTRL Optimization:** $\approx$ **25ms** (Secondary bottleneck).
>
> This breakdown clearly identifies the sources of latency, directly motivating our proposed efficiency variants: **LightCoSteer** is designed to mitigate the optimization bottleneck (d) by simplifying the algorithmic steps, while **AdaCoSteer** addresses the transmission bottleneck (b & e) by adaptively reducing communication rounds. We once again thank you for suggesting this detailed breakdown, as it has been instrumental in clarifying our system workflow and performance characteristics.
>
> ---
>
> **Regarding the intuition behind Iterative FTRL:**
>
> Your intuition is entirely correct! Fundamentally, CoSteer formulates the decoding step as a constrained optimization problem: the goal is to maximize the personalization utility derived from the SLM delta while minimizing the KL divergence from the base LLM to preserve general fluency. The FTRL algorithm is inherently designed to solve such problems through iterative convergence. LightCoSteer (T=1) essentially performs a single gradient step; while efficient, this acts as a linear approximation that may overshoot the safety constraints or under-utilize the steering signal. In contrast, running the process for T=20 allows the algorithm to function as intended—iteratively navigating the probability simplex to settle at a stable equilibrium. This refinement enables CoSteer to locate the precise "sweet spot" that maximizes the personalization delta while strictly adhering to the KL constraint, resulting in the superior generation quality observed in our experiments.
>
> ---
>
> We hope our response have addressed your concerns and remain available for any further questions.

---

### Official Review · Reviewer_7ke3 · 2025-10-26

**Soundness:** 3
**Presentation:** 3
**Contribution:** 2
**Rating:** 6
**Confidence:** 4

**Summary:**

This paper introduces CoSteer, a collaborative framework for personalized text generation that aims to protect user privacy. It addresses a specific and practical scenario where a powerful cloud-based LLM needs to be personalized but cannot be given direct access to a user's sensitive local data (e.g., profiles, interaction history). The core idea is to leverage a smaller, on-device SLM which can access this private context. This local SLM computes a "delta steering" signal, which is the logit difference between its context-aware and context-agnostic outputs. This privacy-safe delta signal is then used to guide the decoding process of the remote cloud LLM, aligning its generation with the user's personal context without that context ever leaving the device. The entire process is formulated as an online optimization problem solved at decoding time.

**Strengths:**

- The paper identifies and tackles an interesting, intuitive, and increasingly relevant problem: how to balance the need for high-quality personalization from powerful cloud models with the critical and non-negotiable requirement of user privacy.

- The proposed CoSteer framework is well-motivated and its core mechanism is straightforward to understand. The idea of using a local model to compute a "delta" to steer a remote model is an elegant solution to this problem.

- The experimental evaluation is comprehensive, testing the framework's effectiveness across multiple personalized generation tasks and different model pairs.

- I appreciate the detailed discussion section (Section 5), which proactively explores practical challenges like robustness to noisy context and collaboration between different model architectures.

**Weaknesses:**

- My primary concern is the paper's limited technical novelty. The contribution is almost entirely in the problem setup and framework design.

- The core optimization algorithm, which is central to the method's implementation, appears to be adopted directly from a previous work (Zhang et al., 2025b), specifically the use of FTRL for decoding-time alignment.

- This lack of technical innovation places a very heavy burden on the novelty of the scenario itself. If this collaborative, delta-steering setup is not demonstrably practical or is perceived as niche, the overall contribution of the paper feels minor.

**Questions:**

- My main question is about the practical grounding of this work. Could the authors comment on or provide any existing examples—whether from public applications or industrial settings they are aware of—that currently use this specific paradigm? That is, a setting where users possess sensitive local information they cannot transmit, and where logit-based signals are used as the medium for collaborative personalization? Evidence of real-world application would significantly strengthen the paper's motivation.

- Thinking about the discussions on noise robustness (5.1) and cross-architecture collaboration (5.2), I wonder if this "Delta Steering" signal could be used for aggregation. For example, could you have multiple different local models compute their respective delta signals, which are then aggregated (e.g., through voting or averaging) to create a single, more robust steering vector that could be less susceptible to the noise or bias of any single local model?

---

> ### Author Response · Authors · 2025-11-20
>
> Dear Reviewer 7ke3,
>
> Thanks for your insightful review and for recognizing the problem we tackle as "**interesting, intuitive, and increasingly relevant**" and our CoSteer framework as an "**elegant solution**." We are also encouraged by your appreciation of our "**comprehensive**" evaluation and the "**detailed discussion**" on practical challenges. Below, we provide point-by-point responses to address your concerns.
>
> ---
>
> **Regarding the concern about technical contribution:**
>
> We respectfully suggest that our technical contribution is multi-faceted, extending beyond the choice of the optimization solver:
> * **Novel Collaborative Framework:** As you correctly pointed out, we shift the paradigm from single-model self-correction to heterogeneous cross-model collaboration. Validating that a tiny, local SLM can effectively steer a massive, remote LLM represents a unique system-level innovation that fundamentally resolves the privacy-utility conflict.
> * **Algorithmic Application:** We openly acknowledge that, similar to previous work (Zhang et al., 2025b), we adopt the FTRL algorithm as the optimizer. We have explicitly cited and acknowledged this work throughout our manuscript. Our contribution here lies in designing a specific **utility function** tailored to the unique constraints of our collaborative scenario, leveraging local deltas to coordinate distributed models rather than focusing on the optimization algorithm itself.
> * **Scenario-Specific Innovations:** To make this framework practically viable in real-world edge environments, we introduced distinct technical contributions: **AdaCoSteer**, which utilizes token confidence for adaptive early termination to reduce communication overhead, and **Byte-Level Fusion**, a vocabulary-agnostic strategy that technically resolves tokenizer mismatches to enable cross-architecture collaboration.
>
> We hope this breakdown clearly articulates the scope of our technical novelty and contributions.
>
> ---
>
> **Regarding the practical grounding and real-world applications:**
>
> We appreciate this question as it touches on the core motivation of our work. We wish to clarify that CoSteer is not a hypothetical construct but stems from a **joint research project** with a leading global provider of ICT infrastructure and smart devices (anonymized for double-blind review). Currently, most commercial personal assistants rely heavily on direct Cloud API calls, often overlooking privacy concerns. While there is a trend toward deploying on-device small models (SLMs), their limited capability creates a bottleneck. Consequently, a critical "vacuum" exists in the current landscape. CoSteer was explicitly designed to fill this gap by operating in conjunction with a gating mechanism: this system intelligently routes tasks based on their nature—handling simple requests locally with SLMs, sending public complex queries to standard Cloud APIs, and selectively activating CoSteer only for tasks that demand both complex reasoning and strict privacy. Furthermore, token-level collaboration was selected as a design choice that effectively balances effectiveness and efficiency. The practical viability of this specific approach is further evidenced by the fact that **related patents have been filed** by our industrial partners based on this framework, confirming that this architecture is viewed as a feasible and valuable direction for next-generation privacy-preserving AI assistants.
>
> ---
>
> **Regarding the potential for delta aggregation:**
>
> We find this suggestion incredibly interesting and insightful. To empirically validate this idea, we conducted an exploratory experiment on the WebQA dataset using Qwen3-8B with Qwen3-0.6B. We compared a centralized approach (concatenating all context documents for a single SLM) against a distributed approach where documents are split among multiple SLMs, whose delta signals are subsequently aggregated via either simple averaging or **entropy-based adaptive weighting**. In the entropy-based strategy, we calculate the prediction entropy for each local SLM and compute the aggregation weights by applying the Softmax function to the negative entropy values, thereby assigning higher importance to models with lower uncertainty. The results are shown below:
>
> |Method|Score|
> |:-|:-|
> |SLM w/o|18.26|
> |SLM w/|39.42|
> |LLM w/o|46.95|
> |Single SLM (Concat.)|49.66|
> |Multi-SLM (Avg.)|48.36|
> |**Multi-SLM (Entropy)**|**50.25**|
> |*LLM w/*|*55.27*|
>
> The results are highly encouraging: while simple averaging (48.36) slightly underperformed the centralized baseline, the **entropy-based adaptive merging** achieved the best performance (50.25). This confirms that parallelizing small models for distributed context processing, coupled with intelligent collaborative steering, is indeed a highly promising direction for future research.
>
> ---
>
> We sincerely thank you for your insightful feedback. We hope our response have addressed your concerns and remain available for any further questions.

---

### Official Review · Reviewer_Yob3 · 2025-10-29

**Soundness:** 3
**Presentation:** 3
**Contribution:** 2
**Rating:** 6
**Confidence:** 3

**Summary:**

This paper develops CoSteer, a framework that allows for personalization at decoding-time by using the difference in logits between a personal and general-purpose small language model to steer cloud-based LLMs. They show through experiments that this approach outperforms the small general-purpose and personal language models and the large general-purpose LLM.

**Strengths:**

- The approach seems like a simple way to leverage the strengths of both personalized SLMs and general LLMs.
- The problem being solved is interesting and relevant for generating good-quality personalized outputs while keeping private information local.
- The paper provides thorough experimental results in a variety of settings, including different SLM-LLM combinations and hyperparameter ablations.

**Weaknesses:**

While the experimental section contains many experiments, the paper should further distinguish the approach from other methods.
1. The paper cites Table 3 to explain why their approach is unique, but I am not sure why exactly the constraints from the table are required. In particular, the main reason why Linear Alignment/Context Steering differ from CoSteer is that these models are not weak-to-strong collaborative. However, LA/CS seem to have fairly comparable performance to CoSteer without weak-to-strong collaboration, so it is not clear why this is needed.
2. The baselines are only compared to CoSteer for a single model pair (Qwen7B-1.5B). Section 4.5 would benefit from additional experiments for other pairs of models.
3. The performance of CoSteer in comparison to the SLM and LLM (Table 1 and 6) seems more mixed for Llama 8B-1B, Qwen 8B-0.6B, and Qwen 8B-32B. Could the authors elaborate why this is the case? I think it would be useful to discuss this more in the main paper.

**Questions:**

- The metrics are listed without standard deviations or error bars. Could the authors add these to the paper?
- There are a variety of typos throughout the paper that should be fixed, particularly misspelled words and missing spaces. Also, Section 4.3 lists the last 2 Qwen models in opposite order from the other models. Is this intentional?

---

> ### Author Response · Authors · 2025-11-20
>
> Dear Reviewer Yob3,
>
> We sincerely thank you for your positive assessment and for recognizing our approach as a "**simple way to leverage the strengths**" of both personalized SLMs and general LLMs. We are particularly encouraged by your appreciation of the problem's "**relevance**" and our "**thorough experimental results**." Below, we provide point-by-point responses to address your concerns.
>
> ---
>
> **Regarding the necessity of weak-to-strong collaboration and the comparison with LA/CS:**
>
> We clarify that Table 3 is intended to illustrate CoSteer's unique **technical positioning**—specifically its privacy-preserving, collaborative architecture—rather than to imply it is the only viable solution in every setting. We acknowledge that the performance advantage of CoSteer over local-only methods (LA/CS) appeared marginal in our original Table 2. However, this gap widens significantly in more resource-constrained settings. As detailed in the response below, new experiments with a tiny Qwen3-0.6B or Llama-1B show CoSteer outperforming LA/CS by a large margin on complex tasks, validating that weak-to-strong collaboration is essential when local models are insufficient.
>
> ---
>
> **Regarding the baseline comparison on additional model pairs:**
>
> We sincerely appreciate this valuable suggestion, which has significantly strengthened the comprehensiveness of our evaluation. We have supplemented **Section 4.5** by conducting experiments across all five model pairs, with detailed results provided in **Table 9 on Page 21** of the revised Appendix. These results confirm that CoSteer consistently outperforms the baselines in the vast majority of scenarios. A particularly striking example is observed in the complex academic abstract writing task using the Qwen3-0.6B model: while the local-only LA/CS approach achieves a ROUGE-1 score of only **31.42** (even below SLM w/o baseline), CoSteer delivers an absolute improvement of over **10 points** (exceeding a 30% relative gain). These expanded findings provide compelling evidence of our method's effectiveness, especially in resource-constrained settings.
>
> ---
>
> **Regarding the performance variations in specific model pairs:**
>
> We sincerely thank you for this professional suggestion. We agree that analyzing the performance nuances across different model pairs significantly enhances the paper's depth and clarity. Accordingly, we have rewritten the "Overall Performance" paragraph in **Section 4.4** of the main text to explicitly discuss these model-specific characteristics (e.g., Llama's stylistic bias and CoSteer's regularization effect). The revised content is as follows:
>
> > "While we observe consistent and robust improvements on standard model pairs (e.g., Qwen2.5 32B-7B and 7B-1.5B), the performance on other configurations exhibits interesting nuances related to model characteristics. For instance, results on the Llama 3 pair vary with output length; it excels in concise *Preference Alignment* tasks but shows more moderate gains in long-form generation, likely due to the compact Llama-1B's stylistic bias towards brevity. Similarly, with the ultra-compact Qwen3-0.6B, while we see strong benefits in complex personalized writing, gains in simpler alignment tasks are less pronounced. We attribute this to CoSteer's regularization mechanism, which conservatively limits steering intensity when the capacity gap is extreme, effectively preventing the tiny SLM from degrading the LLM's general coherence."
>
> We hope this revision provides the clarification you were looking for regarding the mixed results.
>
> ---
>
> **Regarding the inclusion of standard deviations:**
>
> We appreciate this suggestion, as providing variance metrics indeed strengthens the reliability of our results. We have updated **Table 7 on Page 18** of the revised appendix to include both standard deviations and the results of paired t-tests to demonstrate statistical significance. We must admit that the table has become quite "compact" (and perhaps a bit crowded!), but we believe the added statistical rigor provides a much clearer picture of the performance stability.
>
> ---
>
> **Regarding typos and formatting inconsistencies:**
>
> We sincerely thank you for your meticulous attention to detail. We clarify that the inconsistent ordering of the last two Qwen models in Section 4.3 was an unintentional oversight, and we have corrected it to align with the format of the other pairs. Furthermore, we have conducted a thorough proofreading of the entire manuscript, correcting spelling errors, inserting missing spaces, and standardizing the naming conventions for datasets and methods to ensure a polished presentation.
>
> ---
>
> All revisions are highlighted in blue in the updated PDF. We sincerely thank you for your constructive feedback, which has greatly improved our work. We hope our response have addressed your concerns. Please let us know if you have any further questions.

---

### Official Review · Reviewer_WxRf · 2025-10-30

**Soundness:** 2
**Presentation:** 3
**Contribution:** 3
**Rating:** 4
**Confidence:** 4

**Summary:**

This work introduces CoSteer, a framework that enables real-time, privacy-preserving personalization of LLMs by collaborating with small local models. It steers LLM logits using locally computed delta signals from personal-context-aware small models, achieving strong personalized generation without fine-tuning or direct data leakage, validated across diverse tasks and model scales.

**Strengths:**

+ The problem of enabling cloud LLM to be aware of local user data without direct data access is well-motivated.
+ CoSteer introduces a unique collaborative decoding-time personalization framework that enables real-time adaptation using local delta steering without requiring fine-tuning or directly exposing sensitive user data.
+ Extensive experiments across multiple datasets and model scales demonstrate that CoSteer improves personalized text generation while maintaining privacy and efficiency comparable to non-personalized cloud LLM.

**Weaknesses:**

- The core idea builds on existing context steering methods (He et al., 2025), mainly extending them to cloud-edge collaboration, offering limited theoretical or algorithmic innovation.
- Experimental results in Table 2 show only slight improvements over strong personalized baselines.
- Most evaluated datasets are mobile-centric, while CoSteer’s target use case emphasizes cloud LLM serving with personalized requirement.
- The paper lacks a formal theoretical assessment of whether sharing SLM logit differences could indirectly expose sensitive personal information to the cloud.

**Questions:**

Please see Weaknesses.

---

> ### Author Response · Authors · 2025-11-20
>
> Dear Reviewer WxRf,
>
> We sincerely thank you for the constructive review and for recognizing our problem setting as "**well-motivated**" and our framework as "**unique**." We are particularly encouraged by your acknowledgement of our "**extensive experiments**," which validate CoSteer's effectiveness in balancing personalization, privacy, and efficiency. Below, we provide point-by-point responses to address your concerns.
>
> ---
>
> **Regarding the distinction from existing context steering models:**
>
> While previous methods rely on a single model to guide its own generation, our work fundamentally differs by demonstrating that a **tiny, local** SLM can effectively steer a **different, massive** remote LLM. This verifies a novel "weak-to-strong" capability where a small model complements a large one, rather than simple self-improvement. To support this heterogeneous setting, we further introduced specific innovations not needed in single-model approaches: **AdaCoSteer** minimizes communication overhead via confidence-based adaptive termination, and **Byte-Level Fusion** resolves tokenizer mismatches to enable cross-architecture collaboration.
>
> ---
>
> **Regarding the magnitude of improvements in Table 2:**
>
> First, we emphasize that baselines like SFT and Proxy-tuning require computationally expensive training for each user, which is often impractical for real-world on-device deployment due to privacy and storage constraints; we included them primarily for rigorous academic benchmarking. In contrast, CoSteer achieves these results in a completely **tuning-free** manner. Second, while the margins in the specific model pair shown in Table 2 are narrower, we have extended this comparison to all five model pairs in Section 4.5 of the revision. The results demonstrate that CoSteer consistently outperforms these baselines in the vast majority of settings. Notably, in scenarios using highly compact SLMs (e.g., Llama-1B, Qwen-0.6B), CoSteer yields substantial gains, achieving up to a **10-point absolute improvement** (over 30% relative gain) in F1 scores compared to local SLM steering (please refer to the updated **Table 9 in page 21** of the revised PDF).
>
> ---
>
> **Regarding the nature of the evaluated datasets:**
>
> We respectfully wish to clarify that our selection aims to cover a broad spectrum of personal AI agent applications where Cloud LLMs provide significant value. Specifically, **Cogenesis** covers daily assistance like emails and notifications, **LongLaMP** focuses on complex content creation such as abstracts, product reviews, and blog posts, and the four alignment datasets address general preference instruction following. While these requests typically originate from user devices, the complexity of generating high-quality, coherent long-form text often exceeds the capabilities of local SLMs, necessitating the superior reasoning and world knowledge of Cloud LLMs—precisely the gap CoSteer is designed to bridge. If we have misunderstood your definition of "mobile-centric," or if you have specific suggestions for datasets that better represent "cloud LLM serving" in your view, we would warmly welcome your recommendations to further strengthen our evaluation.
>
> ---
>
> **Regarding the privacy assessment of logit differences:**
>
> We respectfully wish to clarify a potential misunderstanding regarding our framework's data flow. In CoSteer, the computation of SLM logit differences (deltas) and the subsequent fusion are performed entirely **locally** on the edge device. Consequently, the cloud server **never** receives the raw logit differences or gradients; it only receives the final, discrete token sampled after local fusion. We fully agree with your insight that any information exchange carries potential risks, which is precisely why we designed this local-execution architecture. Since the final token selection happens on-device, CoSteer naturally supports the integration of local post-processing safeguards (e.g., PII filters) to inspect the fused token. If a token implies a privacy risk, the local system can preemptively halt transmission or mask the content, ensuring sensitive data remains secure.
>
> ---
>
> Once again, we thank you for your valuable feedback and hope that our response and the additional experimental results have satisfactorily addressed your concerns. Please let us know if you have any further questions or require additional details; we would be more than happy to provide further clarifications.

---

> > ### Comment · Reviewer_WxRf · 2025-11-27
> >
> > Thanks for the authors' further clarifications. My major concerns have been well addressed. Please incorporate the clarifications in the revised manuscript to better express the key standpoint of this work. I have increased my rating in correspondence.

---

> > > ### Author Response · Authors · 2025-12-01
> > >
> > > Dear Reviewer WxRf,
> > >
> > > We sincerely thank you for your prompt feedback and for confirming that your major concerns have been well addressed. We are greatly encouraged by your recognition of our clarifications and your decision to **increase the rating** to a positive score.
> > >
> > > Following your valuable suggestion, we have incorporated these critical clarifications into the revised manuscript to better articulate the key standpoints of our work:
> > >
> > > * **Privacy Assessment:** We have explicitly clarified the local-only data flow and potential post-processing safeguards in **Section 3.3** to address privacy concerns formally.
> > > * **Methodological Distinction:** We have expanded **Appendix A.1** to clearly delineate the paradigm shift of CoSteer from existing single-model context steering methods.
> > > * **Dataset Rationale:** We have revised **Appendix B.5** to detail how our selected datasets represent complex cloud-serving scenarios rather than simple mobile tasks.
> > >
> > > We believe these revisions, inspired by your insightful comments, have significantly strengthened the quality and clarity of our paper. Thank you once again for your time and constructive support.

---

### Author Response · Authors · 2025-12-01
**Summary of Rebuttal and Responses**

Dear AC and Reviewers,

We sincerely thank you for your constructive feedback. We are encouraged that reviewers unanimously recognized our problem as "**well-motivated**" and "**critical**," and the CoSteer framework as "**unique**" and "**elegant**," supported by "**comprehensive**" and "**thorough**" experiments.

Notably, during the discussion period, **Reviewer WxRf** explicitly confirmed that our clarifications addressed the concerns and **raised the rating to a positive score** prior to the leakage incident.

Below, we summarize our responses to the common concerns raised during the review process, followed by a summary of specific issues raised by each reviewer.

---

## Response to Common Concerns

**1. Clarification on Technical Positioning:**
We have revised the manuscript to more explicitly articulate our technical positioning, a distinction positively acknowledged by Reviewer WxRf. CoSteer offers a multi-faceted contribution beyond standard inference-time steering:
* **Paradigm Shift:** Validating a novel "weak-to-strong" capability where a tiny, local SLM effectively steers a massive, heterogeneous remote LLM.
* **Formulation:** Repurposing online learning for distributed, privacy-preserving coordination via a novel utility function based on local logit deltas.
* **Innovations:** Introducing AdaCoSteer and Byte-Level Fusion to resolve specific latency and compatibility bottlenecks unique to this architecture.

**2. Expanded Evaluation:**
We expanded our comparative evaluation against local-only methods to all five model pairs (Table 9). Results confirm CoSteer's **consistent superiority** across all configurations. Notably, in resource-constrained settings with tiny models (e.g., Qwen-0.6B), CoSteer outperforms local baselines by **over 10 points** (F1 score), decisively validating the necessity of our collaborative framework.

---

## Responses to Specific Reviewer Issues

* **Reviewer WxRf (Rating Increased):**
    * **Issues:** Perceived privacy risk of logit uploading; Applicability of datasets.
    * **Response:** Clarified that logits are never uploaded (local-only fusion), correcting the **misunderstanding** regarding data flow. Justified dataset suitability for cloud scenarios. The reviewer acknowledged these clarifications and **raised their rating** before.

* **Reviewer Yob3:**
    * **Issues:** Elaborate performance variations; Missing standard deviations; typos.
    * **Response:** Revised Section 4.4 to attribute variations to specific model characteristics and regularization effects. Added standard deviations to Table 7 and fixed typos.

* **Reviewer 7ke3:**
    * **Issues:** Practical grounding; Potential for delta aggregation.
    * **Response:** Clarified the framework's grounding in real-world applications (supported by patent filings).
    Conducted additional exploratory aggregation experiments per suggestion, confirming its future potential.

* **Reviewer QsZr:**
    * **Issues:** Perceived latency burden of local inference; Intuition behind Iterative FTRL.
    * **Response:** Corrected the **misunderstanding** regarding sequential processing; clarified that local inference is masked by network latency via asynchronous pipelining and batching, having **no impact** on the critical path. Added a detailed breakdown (Figure 3) to illustrate this workflow. Explained that iterative FTRL ensures precise convergence compared to linear approximations.

---

We have incorporated all clarifications into the revised manuscript. We believe these revisions have comprehensively addressed all concerns raised during review. Thanks again for your valuable time!

---

### Meta-Review · Area_Chair_fpdX · 2026-01-04

**Summary:**

The paper received two scores marginally below acceptance and two marginally above. Reviewers raised several concerns, including the novelty of the method (which mainly extends the context steering approach to cloud-edge collaboration and is not sufficiently unique), the choice of experimental baselines, and the lack of theoretical analysis. After reviewing the authors’ responses, many of the issues raised by the reviewers remain unclear. The paper requires improvement before it can be considered for acceptance.

**Reviewer Concerns:**

The rebuttal does not provide sufficient explanation of why this approach is novel, as three out of four reviewers raised this concern.

**Reviewer Scores:**

Based on the rebuttal, it is unlikely that the reviewers will revise their scores to positive ratings.

---

### Decision · Program_Chairs · 2026-01-26

Reject